behaviour

vocal diversity, *Urocitellus*, *Spermophilus*, alarm call, acoustic evolution, rodent

**Author for correspondence:**
Lucie Hambálková
e-mail: hambalkoval@fld.czu.cz

# Individual-based acoustic variation of the alarm calls in the long-tailed ground squirrel

Denis Goncharov[1,2], Richard Policht[1],
Lucie Hambálková[1], Viktor Salovarov[2]
and Vlastimil Hart[1]

[1]Department of Game Management and Wildlife Biology, Faculty of Forestry and Wood Sciences, Czech University of Life Sciences Prague, Kamýcká 129, 165 00 Praha 6, Czech Republic
[2]Irkutsk State Agrarian University, 59 Timiryazev St, Irkutsk 664038, Russia

LH, 0000-0002-0025-9325

Based on their phylogenetic position, Nearctic ground squirrels are closest relatives to the long-tailed ground squirrel *Urocitellus undulates* even though it has Palaearctic distribution. We aimed to investigate the variability of alarm calls of the long-tailed ground squirrel to test the individual variation in alarm calls. This species is known to produce two types of alarm calls: whistle alarms and wideband calls. Although ground squirrels are a model group for the study of vocal individuality, this phenomenon has not yet been studied in a species producing two such completely different types of alarms. Most of ground squirrel species produce either whistle or wideband alarms and this species represents a unique model for testing the degree of individual variability depending on completely different acoustic structures. We analysed 269 whistle alarms produced by 13 individuals and 591 wideband alarms from 25 individuals at the western part of Lake Baikal. A discriminant function analysis (DFA) assigned 93.5% (88.9%, cross-validated result) of whistle alarms to the correct individual and 91.4% (84%) of wideband alarms. This is the first evidence of individual variation in wideband alarms compared with whistle alarms and occurrence of vocal individuality in two warning signals of a completely different acoustic structure produced by a ground squirrel.

## 1. Introduction

Vocalization in both Eurasian [1–7] and North American ground squirrels [8–12] has been intensively studied, especially

during the last decade. Even though the long-tailed ground squirrel *Urocitellus undulatus* is a species with a Palaearctic distribution, its phylogenetic position (based on mitochondrial cytochrome b sequences) has been found within the Nearctic clade, and thus Nearctic ground squirrels are this species' closest relatives in comparison with other ground squirrels from the Palaearctic region [13]. The species range includes eastern Kazakhstan, southern Siberia, Transbaikalia, Yakutia (Russia), northern Mongolia and two provinces in China (Xinjiang and Heilungjiang) [14]. The long-tailed ground squirrel (*U. undulatus*) prefers short-grass steppes close to water with a thin chernozem layer and tolerates only low-density bush cover [15]. Two species (*U. undulatus* and *U. parryi*) are considered to have resulted from the more recent east–west crossings of the Bering land bridge [16]. Generic revision of the genus *Spermophilus* revealed eight morphologically distinctive genera, of which only *Spermophilus sensu stricto* is restricted to Eurasia and others have a North American distribution [14]. Of the two species of the genus *Urocitellus*, *U. parryi* has a mostly North American distribution extending to Alaska and Siberia, while *U. undulatus* inhabits the eastern Palaearctic only [14].

Variation in alarm calls of Eurasian ground squirrels has been studied in the speckled ground squirrel *Spermophilus suslicus* [6,17–20], yellow ground squirrel *S. fulvus* [6,17,19,21,22], European ground squirrel *S. citellus* [1–3,5,19,23,24], Taurus ground squirrel *S. taurensis* [1–3,23], Anatolian ground squirrel *S. xanthoprymnus* [3,23,25], red-cheeked ground squirrel *S. erythrogenys* [5], russet ground squirrel *S. major* [26,27], Caucasian mountain *S. musicus* [25], little ground squirrel *S. pygmaeus* [25,28], *S. alaschanicus* [25], Arctic ground squirrel *Urocitellus parryii* [25], grey marmot *Marmota baibacina* [29,30] and Steppe marmot *Marmota bobak* [31]. The basic description of the alarm calls produced by ground squirrels inhabiting the Russian and Asian area included comparative study containing the following species: *Urocitellus undulatus*, U. *parryii*, *Spermophilus xanthopyrmnus*, S. *musicus*, S. *pygmaeus*, S. *alaschanicus*, S. *suslicus*, S. *relictus*, S. *citellus*, S. *erythrogenys*, S. *major*, S. *fulvus* and S. *rally* [30,32]. Results from studies up to now present various levels of distinction in alarm calls of these studied species. There is significant variation in alarm calls shown depending on species [1,2,19,23], sex [6], age [6] and between individuals [2,6,18,20,22,24]. The most striking feature in alarm vocalization of these ground squirrels is that even though their calls constitute a tonal call of very simple structure, they exhibit a high degree of individual variation, which means that each alarm call can be assigned to a specific individual with a high degree of certainty, often within the range 90–100%. The ability to recognize the calling of different individuals may increase the individual fitness of a signal recipient by adjusting its response based on the reliability of the signal. Such a recipient can save potential energetic costs by reducing its response towards unreliable signallers [33,34].

We aimed to investigate the variability of alarm calls of the long-tailed ground squirrel to test the individual variation in alarm calls. This species is known to produce two types of alarm calls: whistle alarms and wideband calls. Although ground squirrels are a model group for the study of vocal individuality, this phenomenon has not yet been studied in a species producing two such completely different types of alarms. The vast majority of ground squirrel species produce either whistle alarms or wideband alarms and this species represents a unique model for testing the degree of individual variability depending on completely different acoustic structures.

# 2. Methods

## 2.1. Study sites and animals

Alarm calls of the long-tailed ground squirrel were recorded in July 2016. In total, 14 natural colonies were found in the area of the Sarma River delta, west of Olkhon Island in the western part of Lake Baikal near to the villages of Sarma (53.1022294° N, 106.8334850° E) and Kurma (53.1799978° N, 106.9654067° E). These locations are at the foot of the Baikal Mountains in Baikal National Park in the Russian Federation. The terrain at both sites is stone and sandy. The elevation ranges from 450 to 500 m above sea level. The delta of the river creates a mosaic of water-filled meadows and dry steppes. The tree cover comprises mainly willow (*Salix* spp.), and also larch (*Larix decidua*) or pine (*Pinus* spp.). The herbaceous vegetation is dominated by grasses, accompanied by meadow blossom. Sparse settlements and free cattle grazing characterize the Sarma village; ground squirrels are not exposed to humans so often here. Kurma village is a growing tourist resort and ground squirrel encounters with humans occur on a daily basis.

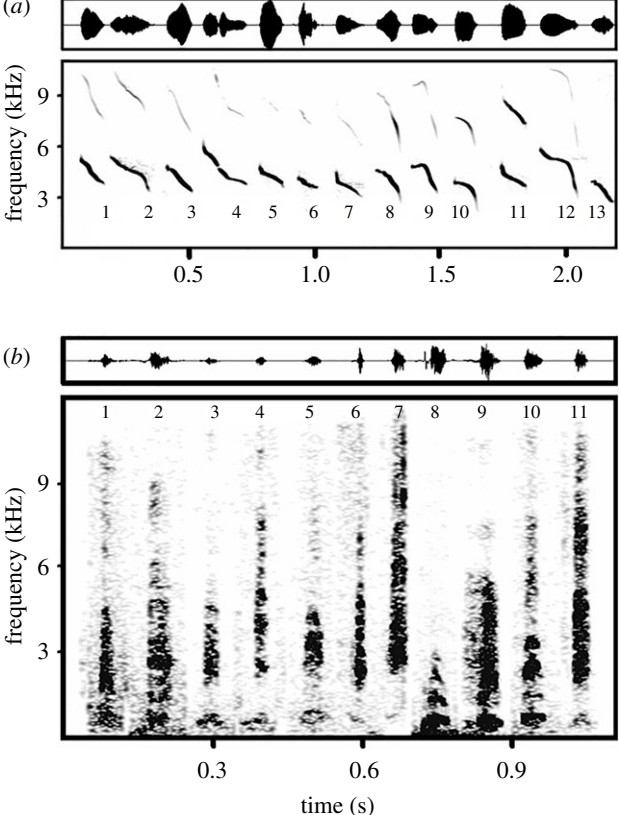

**Figure 1.** Spectrograms and oscilograms of (*a*) whistles from 13 different individuals and of (*b*) wideband alarms of 11 different individuals. Number of individual corresponds to the number assigned to each individual in the scatterplot. The spectrogram (*a*) was created with the following parameters: sampling rate 22.05 kHz, FFT 512, frame 100%, overlap 57.5%. The spectrogram (*b*) was created with the following parameters: sampling rate 24 kHz, FFT 1024, frame 50% and overlap 96.87%.

## 2.2. Recording

During pilot sampling, we tried to record alarm calls of live-trapped individuals according to methods used on other ground squirrel species [1,2] but they did not vocalize at all. Similar results were seen in attempts with live-trapped animals that were placed individually in wire-mesh hutches of 80 × 80 × 80 cm [22]. We therefore decided to record free-running ground squirrels only. Each individual was recorded only once, and when we wanted to record more individuals at the same locality, each researcher selected a focal individual from within the colony and followed them closely to be sure of their identity. These focal individuals were selected from different parts of the colony to be sure with their identity. We categorized two age categories based on the body size: adults and subadults. Subadults were clearly smaller in size, an estimated half the size of adults. The vocalizations were recorded at distances from 3 to 12 m. Duration of each recording ranged from 3 to 10 min, depending on the duration of the calling behaviour of the subject. When we met an individual or a group where at least one group member was calling, we stopped and started recording in a sitting position. Each recorded individual used only one type of call, either an alarm whistle or a wideband signal. The use of the alarm type does not appear to be affected by a context that was approximately constant. We included in the final analysis only calls of those individuals in whom it was possible to observe communication behaviour when the opening of the mouth during the calling could be seen.

Alarm calls were recorded with Olympus Linear PCM LS-5 and ZOOM H5 digital audio recorders in combination with a Sennheiser ME 66 directional microphone (frequency response 20 Hz–20 kHz ± 2.5 dB) equipped with a K6 powering module. Recordings were saved in .wav format (48 kHz sampling rate and 16-bit sample size). Long-tailed ground squirrels produce two types of alarm calls: whistle alarms (figure 1*a*) and wideband alarms (figure 1*b*).

## 2.3. Whistle alarms

We analysed 269 whistle alarms of the highest quality produced by 13 individuals (20.7 ± 7.4 calls, mean ± s.d.). We included only calls with a high signal-to-noise ratio, not disturbed with wind

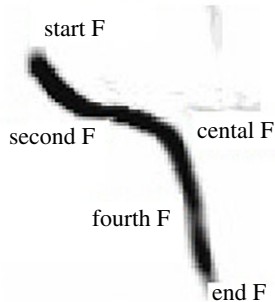

**Figure 2.** Measured points on the spectrograms for calculation of acoustical parameters quantifying the fundamental frequency.

and not overlapping with calls of other individuals or background noises. From these calls, we selected up to 32 calls per individual. In the case of a larger number of calls, we selected them randomly across all recorded series. The recordings were analysed using Avisoft SASLab Pro software (Avisoft Bioacoustics e.K., Schönfließer Str. 83, 16 548 Glienicke/Nordbahn, Germany). Sampling frequency was converted to 22 050 Hz and selected calls were manually labelled. We used spectrograms with the following parameters: Hamming window, FFT-length 1024, frame size 50% and overlap 96.87%. When background noise was present, it was filtered out using both high-pass (1.3–2.9 kHz) and low-pass filters (5.2–7.7 kHz). These filters show the range both for the lower- and upper-frequency limits. When surrounding noise was present, we filtered that noise which was out of the signal, e.g. when fundamental frequency reached up to 6.65 kHz, we filtered out the frequency higher than 7 kHz.

We measured 11 acoustic variables quantifying a fundamental frequency of single whistle call: duration, five frequency parameters (figure 2) using automatic measurements at five regular intervals of fundamental frequency duration of one whistle call (start frequency, second frequency, central frequency, fourth frequency, end frequency) that divided each whistle into four sections in which we measured frequency modulation as follows: first frequency modulation (start frequency minus second frequency), second frequency modulation (second frequency minus central frequency), third frequency modulation (central frequency minus fourth frequency) and fourth frequency modulation (fourth frequency minus end frequency), and frequency range (start frequency minus end frequency).

## 2.4. Wideband alarms

We selected 591 wideband alarms (23.7 ± 6.0 calls) of the highest quality from 25 individuals. Similarly to whistle alarms, we included only calls with a high signal-to-noise ratio, not overlapped with other calls or sounds. Signals where there was no clear lower frequency limit of the signal due to overlap with low-frequency noise were not selected. From selected calls, we randomly selected up to 34 calls per individual across all recorded series.

Sampling frequency was converted to 24 kHz and the recordings were filtered using a high-pass filter (1.6 kHz) to remove the background noise, when it was present, to increase the signal-to-noise ratio. The recordings were analysed using Raven Pro 1.5 software. Wideband alarms were manually labelled on the spectrograms set with the following parameters: Hamming window, FFT-length 512, frame size 50% and overlap 94%. Spectrogram measurements provided frequency parameters and the amplitude curve provided temporal parameters. We automatically measured 22 parameters quantifying how the acoustic energy is spread across the frequency spectrum and duration of the signal (table 1).

## 2.5. Statistical analyses

Individuals with a lower number of whistle alarms or wideband alarms ($n < 9$) were not included in the discriminant analysis even though they were included for descriptive statistics. Highly correlated variables were omitted ($r \geq 0.9$). The remaining 16 variables were entered into discrimination function analysis (DFA) using IBM SPSS Statistics 24.0 software (IBM Corp., Armonk, USA). In the case of whistles, all six parameters were entered into DFA. We used a leave-one-out cross-validation

**Table 1.** Descriptions of acoustic parameters measured in Raven Pro 1.5.

| acoustic parameter name | description |
| --- | --- |
| call duration [duration(s)] | duration of the call. |
| first quartile frequency [Q1F(Hz)] | the frequency that divides the selection into two frequency intervals containing 25% and 75% of the energy in the signal. |
| relative first quartile frequency [Q1FRel] | the frequency that divides the selection into two frequency Intervals containing 25% and 75% of the energy in the signal relative to the frequency range of the signal. |
| third quartile frequency [Q3F(Hz)] | the frequency that divides the selection into two frequency intervals containing 75% and 25% of the energy in the signal. |
| relative third quartile frequency [Q3FRel] | the frequency that divides the selection into two frequency intervals containing 75% and 25% of the energy in the signal relative to the frequency range of the signal. |
| aggregate entropy [AggEnt(bits)] | the aggregate entropy measures the disorder in a sound by analysing the energy distribution. Higher entropy values correspond to greater disorder in the sound whereas a pure tone with energy in only one frequency bin would have zero entropy. It corresponds to the overall disorder in the sound. |
| average entropy [AvgEntr(bits)] | this entropy is calculated by finding the entropy for each frame in the signal and then taking the average of these values. |
| max entropy [MaxEntr(bits)] | maximum entropy calculated from each frame. |
| bandwidth 90% [BW90%(Hz)] | the difference between the 5% and 95% frequencies. |
| centre frequency [CentreF(Hz)] | the frequency that divides the selection into two frequency intervals of equal energy. |
| centre time [CentreT(s)] | the point in time at which the selection is divided into two time intervals of equal energy. |
| relative centre time [CentreTRel] | the point in time at which the selection is divided into two time intervals of equal energy relative to the signal duration. |
| frequency 5% [F5%(Hz)] | the frequency that divides the selection into two frequency intervals containing 5% and 95%. |
| relative frequency 5% [F5%Rel] | the frequency that divides the selection into two frequency intervals containing 5% and 95% relative to frequency range. |
| frequency 95% [F95%(Hz)] | the frequency that divides the selection into two frequency intervals containing 95% and 5%. |
| relative frequency 95% [F95%Rel] | the frequency that divides the selection into two frequency intervals containing 95% and 5% relative to frequency range. |
| inter-quartile range [IQRBW(Hz)] | the difference between the first and third quartile frequencies. |
| peak frequency [MaxF(Hz)] | the frequency at which the maximum power occurs. |
| time 5% [T5%(s)] | the time that divides the signal into two time intervals containing 5% and 95%. |

(*Continued.*)

| acoustic parameter name | description |
| --- | --- |
| relative time 5% [T5%Rel] | the time that divides the signal into two time intervals containing 5% and 95% relative to signal duration. |
| time 95% [T95%(s)] | the time that divides the signal into two time intervals containing 95% and 5%. |
| relative time 95% [T95%Rel] | the time that divides the signal into two time intervals containing 95% and 5% relative to signal duration. |

procedure (IBM SPSS Statistics 20) to validate the results of DFA. $N$ refers to the number of individuals; $n$ refers to the number of calls.

To avoid a potential bias in DFA results due to an unbalanced dataset, we randomly selected nine calls per each individual and such balanced datasets entered into each DFA (whistles alarms: $n = 108$ calls, $N = 12$ individuals; wideband alarms: $n = 207$ calls, $N = 23$ individuals). The descriptive statistics were calculated from the average values per each individual. All means were indicated as mean ± s.d.

# 3. Results

## 3.1. Alarm call description

Whistle alarms represent one-syllable calls with a visible modulated fundamental frequency (figure 1a). Duration of whistle alarms ranged from 0.03 to 0.21 s ($0.10 \pm 0.03$, mean ± s.d.). The frequency rapidly decreases and peak frequency always coincides with the fundamental frequency ($f0$), sometimes with visible harmonics, and if so, these visible harmonics are always at a weaker intensity.

Only one individual showed a short, slight increase in the frequency of the initial segment followed by a rapid decrease (see individual 9; figure 1a). Fundamental frequency started at $4811 \pm 641$ Hz and rapidly decreased to $3439 \pm 339$ Hz (table 2).

In comparison with whistle alarms, the second alarm call type is a monosyllabic call with a wideband structure, in which acoustic energy is spread across the frequency spectrum and throughout the signal duration. Signals do not show a fundamental frequency or harmonic components but instead have several emphasized frequency bands (figure 1b). These alarm signals are significantly shorter than whistle alarms (duration: $0.028 \pm 0.007$ s, Mann–Whitney U test: $p < 0.001$). The frequency of the first quartile frequency was $2317 \pm 632$ Hz and the third quartile frequency $3375 \pm 1119$ Hz. Peak frequency was found at $2456 \pm 650$ Hz. Other descriptive statistics are shown in table 2.

## 3.2. Individual variation

### 3.2.1. Whistle alarms

The discrimination analyses considered 108 whistle alarms from 12 individuals. The model included five variables (fourth and second frequency, first, fourth and third frequency modulation). A discriminant function analysis assigned 93.5% (88.9% cross-validated result) of whistle alarms to the correct individual ($N = 12$, $n = 108$, Wilks' lambda = 0.001). The first four significant discriminant functions ($p < 0.001$) had eigenvalues greater than 1 and explained 97.4% of the variation. The first two functions had eigenvalues greater than 5 and explained 86.5% of the variation. The first discrimination function correlated well with the fourth frequency ($r = 0.65$), while the second discrimination function correlated with the third frequency modulation ($r = 0.48$). Five individuals reached the highest possible classification accuracy (100%), four squirrels were classified with an 80–90% result, two animals showed 78% and one individual 56% classification result. Such classification results were much higher in comparison with classification by chance (8.3%). Location of whistle alarms in the space of the first two discriminant functions is shown in figure 3a.

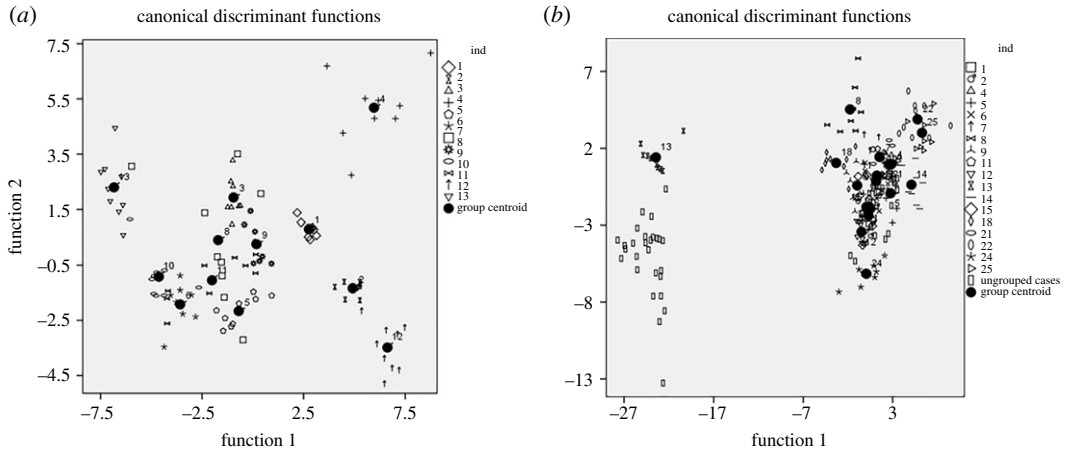

**Figure 3.** Scatterplot on the basis of two first discriminant functions of DFA for (*a*) whistle alarms from 13 individuals and for (*b*) wideband alarms from 25 individuals. Black circles show group centroids.

**Table 2.** Basic descriptive statistics.

| acoustic parameter | mean | s.d. | min | max |
|---|---|---|---|---|
| *whistle alarms* | | | | |
| duration (s) | 0.10 | 0.03 | 0.03 | 0.21 |
| start frequency (Hz) | 4810.9 | 641.1 | 3350.0 | 6650.0 |
| second frequency (Hz) | 4511.0 | 550.3 | 3530.0 | 5980.0 |
| centre frequency (Hz) | 4245.3 | 498.2 | 3380.0 | 5570.0 |
| fourth frequency (Hz) | 3904.8 | 417.8 | 2900.0 | 4950.0 |
| end frequency (Hz) | 3438.7 | 339.0 | 2560.0 | 4540.0 |
| *wideband alarms* | | | | |
| duration (s) | 0.028 | 0.007 | 0.014 | 0.048 |
| first quartile frequency (Hz) | 2316.8 | 631.5 | 281.2 | 3609.4 |
| third quartile frequency (Hz) | 3375.3 | 1118.8 | 421.9 | 7359.4 |
| bandwidth 90% (Hz) | 2984.9 | 1483.8 | 140.6 | 7453.1 |
| centre frequency (Hz) | 2678.8 | 751.6 | 375.0 | 4968.8 |
| frequency 5% (Hz) | 2105.4 | 650.4 | 46.9 | 3000.0 |
| frequency 95% (Hz) | 5090.3 | 1756.5 | 562.5 | 9796.9 |
| peak frequency (Hz) | 2455.5 | 649.6 | 328.1 | 4968.8 |

### 3.2.2. Wideband alarms

We entered 207 wideband alarms from 23 individuals into the following analysis. The model contained nine acoustic variables. The model assigned 91.4% (84%, cross-validated result) of wideband alarms to the correct individual ($N = 23$, $n = 207$, Wilks' lambda less than 0.001). The first five significant discriminant functions ($p < 0.001$) had eigenvalues greater than 1 and explained 96.6% of the variation. The first two functions had eigenvalues greater than 7 and explained 82.8% of the variation. Frequency 5% Rel. mostly correlated with the first discrimination function ($r = 0.88$), and duration with the second discrimination function ($r = 0.63$). Wideband alarms were assigned to the correct caller with a 55.6–100% result. Most individuals ($N = 8$) showed the highest classification accuracy (100%) and five animals were classified with a 60–89% result. Three individuals were classified with 56% success. These results were much higher than a classification by chance (5.6%). Location of wideband alarms in the space of the first two discriminant functions is shown in figure 3*b*.

### 3.2.3. Age effect

Additionally, we tested a potential distinctness between age and did not find a significant difference in any of the acoustic parameters ($0.11 < p < 0.61$; Mann–Whitney U test) in both call types.

## 4. Discussion

The split of Eurasian ground squirrels into two genera—*Spermophilus* and *Urocitellus* [13]—has been shown to be mirrored by vocalization pattern as well. We have described two different types of alarm calls in the long-tailed ground squirrel *Urocitellus undulatus*: (i) a whistle alarm call corresponding with whistle alarms of other ground squirrels inhabiting the Palaearctic area, but also (ii) a wideband alarm call representing a different design of alarm signal (produced both by long-tailed ground squirrel and arctic ground squirrel) has no analogue in any other Eurasian ground squirrel species. The Eurasian species of ground squirrels have wideband calls similar in structure in their vocal repertoire, but they are not used as alarm calls. In contrast with Eurasian species, which are known to produce whistle alarms only, some North American ground squirrels are known to produce both whistle and wideband alarms [19]. In our study, both call types did not differ between adults and subadults. Although young individuals differ significantly in size from adults, the fundamental frequency of alarm whistles does not differ in other studied Palearctic ground squirrels [17,35,36]. According to our results, such phenomenon seems to be valid also for wideband alarm signal. Age information could be concealed in alarm calls of Palearctic ground squirrels [36]. The two different types of alarm calls in North American species are considered as responses to aerial (whistle alarms) and terrestrial predators (wideband alarms) [37,38], but in our study, both alarm types were produced in response to terrestrial subjects as well.

Both whistle and wideband alarm calls were specific at the individual level. Based on our discrimination models, we were able to assign each whistle to the correct individual with 93.5% average success (88.9% based on cross-validated result) and wideband alarms with 91.4% average success (84%, cross-validated results). Such results are consistent with classification outputs of whistle alarms in other ground squirrels: 92% (*Spermophilus suslicus*) [18], 94% (*S. fulvus*) [21], 97% (*S. citellus*) and 95% (*S. taurensis*) [2]. This study is the first to examine individual variation in wideband alarms compared with whistle alarms and occurrence of vocal individuality in two warning signals of a completely different acoustic structure.

## 5. Conclusion

In comparison with other Eurasian species, the long-tailed ground squirrel produces two types of alarms: (i) a whistle alarm, representing a simple signal comparable with whistles of other Eurasian species, even though it showed close similarity to simple whistles of North American species and (ii) a wideband alarm. We found that both signals are individually specific. This is the first evidence of an individually distinct structure in the wideband alarms of any ground squirrel and occurrence of vocal individuality in two warning signals of a completely different acoustic structure. We may speculate that the previous split of Eurasian ground squirrels into the two genera *Spermophilus* and *Urocitellus* based on genetics is supported by vocalization as well.

Ethics. The research was conducted in accordance with the guidelines of the Animal Behaviour Society for the ethical use of animals in research. The study was carried out in accordance with the recommendations in the Guide for Care and Use of Animals of the Czech University of Life Sciences Prague. The Animal Care and Use Committee of the Czech Ministry of the Environment approved the protocol (no.: 4/19). The Institute of Natural Resources Management of Irkutsk State Agrarian University named after A.A. Ezhevsky states that project Bioacoustic research of mammals and birds in the Irkutsk region has been considered to belong to the bilateral cooperation between our countries within the framework of wildlife biology. According to the Russian legislation, this type of scientific projects does not need any special permits or licences. In a more general way, observing, recording, catching, tagging and letting animals go back to the wild nature do not need any permission of the landowner.

Data accessibility. Data are included as electronic supplementary material.

Competing interests. We declare we have no competing interests

Funding. This work was supported by the Internal Grant Agency of the Faculty of Forestry and Wood Sciences, Czech University of Life Sciences Prague, reg. no.: B06/16.

Acknowledgements. Not applicable

Supplementary material. All data generated or analysed during this study are included in this published article or its electronic supplementary material files.

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
