## [Peer Review File · Royal Society Open Science]

Review History

RSOS-200147.R0 (Original submission)

Review form: Reviewer 1

Is the manuscript scientifically sound in its present form?

No

Are the interpretations and conclusions justified by the results?

No

Is the language acceptable?

Yes

Do you have any ethical concerns with this paper?

No

Have you any concerns about statistical analyses in this paper?

Yes

Recommendation?

Major revision is needed (please make suggestions in comments)

Comments to the Author(s)

In this study, the authors investigate the acoustics and individuality of two types of the alarm call in the Eurasian ground squirrel species *Urocitellus undulatus*, closely related phylogenetically to North American ground squirrels. The authors discuss the similarities and differences in the acoustics of the alarm call and compare the individuality of the two calls types. In addition, the authors compare the alarm calls of *Urocitellus undulatus* with the alarm calls of other species of ground squirrels of Eurasia and North America. This study is very interesting for the broad scientific readership working with diversity of alarm calls across taxa.

Overall, the MS has a high scientific value. However, it is not well-written. The collected by the authors excellent material deserves a more accurate writing.

I have a few main concerns to this MS.

First concern. This study advertise the comparative analysis of alarm calls across species.

However, the authors did not include a large body of literature, describing the alarm calls of many species of Eurasian and North American ground squirrels. This ignored body of literature includes the papers, describing in the first time the wideband alarm calls of the long-tailed and the Arctic ground squirrels and tracking the bioacoustical relationships between the Eurasian and North American species, what is the main focus of the MS. These studies should be obligatory considered in the MS. References to these study are provided in the referee comments below.

Second concern. Methods are non-transparent. Remains unclear, how the animals were recorded, how the caller age class was determined, how the calls were selected for analysis and how the acoustic variables of the calls were measured. The authors should obligatory provide the large-scale illustrative spectrograms showing which acoustic variables were measured for each call type. Descriptions of the acoustic variables, used for inter-species comparison of the alarm calls of ground squirrels, are lacking. Remains unclear, from where and how the data introduced by the authors in the cluster-analysis, were obtained, what leads to great doubts in the obtained results.

Third concern is related to the acoustic analysis. The provided by the authors description of the measured acoustic variables does not allow to compare them with those commonly measured in the similar papers. The filtering mode, applied by the authors to the acoustic file, should affect the acoustic structure of both the whistle and the wideband alarm calls. Judging by Figure 2, this procedure just will cut off the lower halves of the wideband alarms. The authors should explain this and probably re-calculate their results without applying this damaging filtering.

Fourth concern is related to statistic analyses. I suggest calculating the average values for each acoustic variable for each individual and then use them for the descriptive statistics. Such sample is more balanced than data pooling, used in the MS. The results of DFA are dependent from the sample size. Probably, that the low value of correct assignment for some individuals is related to the misbalanced call samples. I also found that calculation of DFA classification by chance was made incorrectly and should be re-calculated.

I suggest to change the title, as it does not reflect the content of the MS. I also suggest to add some short information about the results of analyses of individuality in Abstract. The language is overall acceptable (for the exclusion of a few unclear sentences) and was corrected by the professional proofreading with editing services. The revised MS should be then re-reviewed again.

Ethics

Line 22 Please correct typos in the word "regional"

Title

The title is not perfectly clear, what the authors mean under the unique signal? Do you indeed compare individual identity of the alarm call between American and Eurasian squirrels? Please correct the title. Probable variant: "Individual-based acoustic variation of the alarm call in the long-tailed ground squirrel suggests a closer relationship with North American than Eurasian ground squirrels".

Abstract

Abstract says nothing regarding the individual variation, please add you results about individuality in Abstract.

Line 17 Delete “the” before “vocalization”

Line 19 add “ground squirrel” after “species”

Lines 20-21. We found a unique signal, which confirms the closer relationship of Eurasian ground squirrels to North American species, rather than other species from Eurasia. This finding is in concordance with previous molecular results.

If I understood correctly the sense of this sentence, it should be corrected as follows “We found that the alarm call of the long-tailed ground squirrel was distinctive among Eurasian species of ground squirrels but was similar with those of North American species of ground squirrels”
Please re-write.

Lines 22-24 The split of Eurasian ground squirrels into two genera – Spermophilus and Uroditellus – based on genetics is supported by vocalization.

I suggest re-writing to “We discuss that these acoustic data confirm the genetic-based separation of Eurasian ground squirrels to the two genera, Spermophilus and Uroditellus.”

Introduction

Lines 30-31. Vocalization in both Eurasian and North American ground squirrels has been intensively studied, especially during the last decade.

References are necessary at the end of this sentence.

Line 37. The long-tailed ground squirrel species

Unclear, what the authors mean here, one species – the long-tailed ground squirrel of both species of genus Uroditellus (the long-tailed and the Arctic)

Line 45. Such a situation is unique among ground squirrels studied so far.

Delete this sentence

Lines 46-47 The question at hand is whether alarm call in long-tailed squirrel could be a result of phylogenetic correlation with other species.

This sentence is unclear. Do you mean the result of phylogenetic relationship with other species?

Lines 51-52 Correct citations. European - delete 16 and insert 17, Anatolian – change 12 to 13

Lines 49-53. In reality, descriptions of the alarm calls of Spermophilus ground squirrels are available for more number of species. As this study pretends to provide a reviewing comparison, the authors should obligatory include these papers in the MS. Most of these papers are in Russian, however, two of the authors are native Russian speakers, so the reading these papers will not be difficult for them. Most of these studies are rather old and the acoustic analyses are weak, but they cannot be ignored. References are provided below.

Nikolskii, 1979. Species specificity of alarm call in sousliks of (Citellus, Sciuridae) Eurasia. Zool. Zhurnal, v. 58, N 8, p. 1183-1194 and

Nikolskii, 1984. Acoustic signals of mammals in the evolutionary process. Moscow, Nauka.

Description of species-specific alarm calls of ground squirrels of Eurasia *S. citellus*, *S. suslicus*, *S. xanthopyrmnus*, *S. pygmaeus*, *S. alaschianicus*, *S. erythrogenys*, *S. maior*, *S. fulvus*, *S. relictus*. In addition, spectrograms of the alarm calls of *U. undulatus* and *U. parryi* are provided. A

description of the bioacoustical relationships of the two last species with ground squirrels of North America.

Nikolskii et al., 1984. The alarm call in F1 hybrids *Citellus pygmaeus* X *C. suslica* (Sciuridae, Rodentia). *Zool Zhurnal*, v. 63, N 8, p. 1216-1225.

Nikolskii, Starikov, 1997. Variability of alarm call in *Spermophilus maior* and *Spermophilus erythrogenys* (Rodentia, Sciuridae) within contact zone in Kurgan district. *Zool Zhurnal*, v. 76, N 7, p. 845-857.

Nikolskii et al., 2007. Geographical variability of the little ground squirrel (*Spermophilus pygmaeus*): a bioacoustic analysis. *Zool Zhurnal*, v. 86, N 11, p. 1379-1388.

Nikol'skii, 2019. Sex accent and biphonation in the sound signal of ground squirrels (Mammalia, Rodentia). *Doklady Biological Sciences*, 2019, V. 487, p. 119-123. Russian Text in *Doklady Akademii Nauk*, 2019, V. 487, No. 6.

Line 55 Delete "especially". Otherwise, it sounds as that individual-level variation is higher than the species-level variation, what is not the case in ground squirrels.

Line 56. Correct citations. Add 6 and 9

Line 57-58. a whistle sound with simple tonal structure

Please delete "whistle" here. Whereas calls of some rodents indeed represent the aerodynamic whistles by the production mechanism (Riede, 2011, 2013; Pasch et al. 2017; Riede et al. 2017; Riede, Pasch 2020) and in ground squirrels probably too, this production mode has not yet been confirmed for the alarm calls of ground squirrels.

Please re-write this sentence as "tonal calls of very simple structure"

Line 66 "to test the individual variation in alarm calls"

Individual variation was not mentioned in Abstract and appeared in the first time only in the aim. Please add something you findings regarding the individuality of alarm calls in Abstract.

Line 67 strongly species-specific. Please add reference to Nikolskii 1979, 1984

Line 67 supposed to be innate. Please add reference to Matocha, 1975. Vocal communication in ground squirrels, genus *Spermophilus*. PhD thesis, Graduate Faculty of Texas Tech University.

Methods

Line 82 the first study site. Change to Sarma village

Line 91-94. Please indicate the age of animal callers (adults or not) and how they were aged.

Please indicate the distance during recording (from - to). Please indicate the approximate duration of recording. Please indicate what the observer did during the recording, because the behaviour of researcher affect animal vocal activity and producing either the whistle or the wideband alarms. Please indicate, what calls the animals produced: only the whistle alarms, only the wideband alarms or both call types. Please indicate, how many animals you recorded and how they were distributed by the 14 colonies.

Most important - please indicate, by which traits you determine that you record the focal animal rather than the hidden caller. My experience of recording ground squirrels in the field suggests that mistake is highly likely, especially if the silent focal animal is partially hidden or is moving, and the second (caller) is nearby but hidden perfectly from the researcher (by grass, bush etc.). This is critically important for the results on individuality of the alarm calls.

Lines 95-96 "The Animal Care and Use Committee of the Czech Ministry of the Environment approved this research."

Please transfer this sentence in the Ethics section.

Line 102 “Whistles”

Change “Whistles” to “Whistle alarms” throughout. The authors should indicate here that long-tailed ground squirrels produce two types of alarm calls, the whistle alarm and the wideband alarm, and refer to the respective figures with spectrograms.

Line 103 We analysed whistles of the highest quality.

Please provide the number of selected whistle alarms (as you did below for the wideband alarms).

Please provide the criteria of the highest quality (for example, calls with high signal-to noise ratio, not broken with wind, not overlapped with alien calls or noises etc.).

Line 103. Please indicate in detail, how many whistle alarms were selected for analyses. From how many individuals. How many calls per animal (Mean, SD). How calls were selected within individual (from one of from a few series, following each other or not).

Lines 104 and further. software

Please provide the firm, headquarters city and country of producer, this is a common claim of all journals.

Lines 106-107 Background noise was filtered out using both high-pass (1.3–2.9 kHz) and low-pass filters (5.2–7.7 kHz).

The authors should write here, why these frequency ranges were filtered out. Remains unclear, whether such filtering mode affected the important parts of the studied calls. From Table 1 follows, that maximum values of the start fundamental frequency (6.65 kHz) are within the zone which was filtered out.

Lines 107-115

This list of measured acoustic parameters is non-informative. Unclear what the authors indeed measured. What do you mean under “frequency”? The fundamental frequency? Or the automatically measured peak frequency (frequency of maximum amplitude)? The authors should obligatorily provide the figure illustrating all measured acoustic variables.

Line 118 “We selected...”

Please provide criteria for this selection of wideband alarm calls for analysis. Please indicate in detail, how many wideband alarms was selected per animal (Mean, SD). How calls were selected within individual (from one or a few call series, following each other or not).

Lines 119-120 Why 1.6 kHz high-pass filtering was applied for filtration? What was frequency range of the calls? Were meaningful call part damaged due to this filtering? Judging by spectrograms presented on Figure 2, the authors cut-off the lower halves of calls and then only analyses the upper halves. This is very strange and demands a detailed explanation. Moreover, many values of acoustic variables indicated in Table2 are lower 1600 Hz. How it is possible?

Line 125 Reference to Table 2 appears in text earlier then the reference to Table 1. Please correct this.

Line 124 We measured 22 parameters

How these 22 parameters were measured? Manually? Automatically?

Lines 124-125, Table 2. This description of measured parameters in insufficient. Please provide a detailed description of parameters being measured and provide a figure with illustrative spectrogram showing the measured acoustic variables. The used acoustic variables are unclear

and are incompatible with the acoustic variables used in other papers on the alarm calls of ground squirrels, what makes the results of MS incomparable with other published studies. Please, make the descriptions of the acoustic variables clear at least for reviewer - bioacoustic researcher.

For example, 1st Quartile Frequency – this is a part of call spectrum, covering 25% of the total call energy. However, Table 2 describes the 1st Quartile Frequency as "The frequency that divides the selection into two frequency intervals containing 25% and 75% of the energy in the signal". How Q1F differs from Q1F_{rel}, which is determined as "The frequency that divides the selection into two frequency Intervals containing 25% and 75% of the energy in the signal relative to the frequency range of the signal"?

"Maximum entropy calculated from each frame". What is frame and how many frames?

"The difference between the 5% and 95% frequencies". Bandwidth is the width of the frequency band of call spectrum. What you understand under "frequency"? How 1.6 kHz high-pass filtering affects the bandwidth?

Center Frequency "The frequency that divides the selection into two frequency intervals of equal energy". What frequency, fundamental or peak? Or this is the median of call power spectrum?

.... etc.

Line 128. Individuals with a lower number of calls ($n < 7$)

What calls, whistle alarms or wideband alarms?

Line 128-134. The results of a DFA are dependent from the sample size. I would suggest to balance the data set by number of calls per individual, included in each DFA.

Line 137-138. Six acoustic variables (Call duration, Element duration, Start Frequency, End Frequency, Min Frequency and Max Frequency)

It is necessary to provide a detailed description of these acoustic variables. Call duration – was this the duration of a series of repetitive calls? Frequency – was this the fundamental or the peak frequency? How it was measured when the alarm call represented a series of repetitive calls?

Without clear methods, results cannot be understood.

Results

Line 146 and further. Alarm call description

As you analysed evidently different call numbers from different individuals, would be better to calculate the average values of acoustic variable for each individual and then use them in the descriptive statistics in Table 1. Such sample will be more balanced for calculating the averages per species than data pooling, used in MS.

Line 147 tonal whistles

Change to whistle alarms throughout, uniformly with wideband alarms.

Lines 148-149 "calls with a tonal frequency modulated structure formed of one syllable only"
I suggest re-writing to "one-syllable calls with a visible modulated fundamental frequency"

Line 149 mean \pm SD

Transfer this in Statistics section, e.g. "All means were indicated as mean \pm SD".

Line 156 the second type of alarm signal

Please avoid synonyms, moreover "signal" is incorrect term here. I suggest rewriting to "the second alarm call type,"

Line 166 Tonal alarms (Whistles)

Do not use synonyms, change with whistle alarms

Lines 167-168. We excluded the highly correlated 2nd Frequency from all subsequent analysis. In Methods (Lines 131-132) you wrote that all 6 acoustic variables of whistle alarms were included in DFA. Please correct.

Lines 167-179. The discrimination analyses considered 269 whistles from 13 individuals. The results of a DFA are dependent from the sample size. Please provide a table with DFA results, including the number of whistle alarms from each of the 13 individuals and percentages of correct assignment of calls to each individual. Probably, the low value of correct assignment for some individuals is related to misbalanced sample.

Lines 178-179. Such classification results were much higher in comparison to classification by chance (7.7%). Classification by chance was calculated incorrectly. Please use either the method by Solow, 1990 (Ecology, 71:2379-2382), or by Mundry, Sommer, 2007 (Animal Behaviour, 74:965-976)

Lines 182-183. The model contained ten acoustic variables. Again, earlier in Methods (Line 130) is indicated that 16 acoustic variables was included in analysis. Please correct.

Lines 182-192. We entered 591 wideband alarms from 25 individuals into the following analysis. The results of a DFA are dependent from the sample size. Please provide a table with DFA results, providing the numbers of wideband alarms from each of the 25 individuals and percentages of correct assignment to each individual. Probably, the low value of correct assignment for some individuals is related to misbalanced sample.

Lines 191-192. These results were much higher than classification by chance (4%). Classification by chance was calculated incorrectly. Please use either the method by Solow, 1990 (Ecology, 71:2379-2382), or by Mundry, Sommer, 2007 (Animal Behaviour, 74:965-976)

Line 194. Fig. 4
Reference to Fig 3 and description of this figure are lacking in the MS.

Line 200 and further. Species comparison

Unfortunately, this interesting section is written based on incomplete and unclear how obtained data. Accordingly, the results of this section have a negligible meaning. This section should be re-worked and re-calculated perfectly.

First, the authors should provide in Methods a detailed description of the used acoustic variables. For example, what frequency was measured (fundamental or peak?) in the wideband alarms of the long-tailed ground squirrel and how it was measured? And how this measurement agrees with similar measurements in the tonal alarms of calls of other species. I think that the results of Fig 9 about the separate position of the wideband alarm of the long-tailed ground squirrel directly result from the incorrect measurements of "frequency".

Second, remains unclear, from where and how the data introduced by the authors in cluster-analysis were obtained. Were they taken from the papers? Or from the own measurements of calls taken from web-sites? Of how many calls and from how many individuals?

Third, interspecies comparisons of the alarm calls of ground squirrels (Fig. 9 and Additional file 1) are evidently incomplete. Alarms of a few species available from literature (and even presented on Figs 5-8, but not included in Fig 9) should be included. Data of the cue papers on the Arctic ground squirrel are lacking, although they are critically important for comparison of Eurasian and North American ground squirrels (Melchior, 1971, Oecologia, 7:184-190; Nikolskii, 1979, 1984). The Arctic ground squirrel lives in tundra on both sides of Bering Strait and produces both the whistle and wideband alarms, very similar with the respective call types of the long-tailed

ground squirrel. Without inclusion of these data, the picture of uniqueness of alarm calls of long-tailed ground squirrel in distorted.

The missing in MS references on the alarm calls of ground squirrels are available in Matrosova et al. 2012 (cited in MS) and in a new review on vocalizations of ground squirrels (Newar, Bowman, 2020, *Front. Ecol. Evol.*, 2020, v. 8: 00193. doi: 10.3389/fevo.2020.00193), especially in Supplementary to this review.

Lines 203-205 These calls represent tonal calls: monosyllabic whistles (A, D, E), two-syllable whistles (B, C), and multisyllabic chirps (F) and churrs (G).

Whistles, chirps, churrs: this terminology is redundant and only confuses the reader. I suggest re-writing to: "These alarm calls are all tonal calls: monosyllabic (A, D, E), two-syllabic (B, C), and multisyllabic (F, G)."

Lines 205-208 "Calls of North American species (Fig. 6) also represent monosyllabic whistles (A, B, C), two-syllable whistles (D), and multisyllabic chirps (E, F). Similarly, calls of the genera *Otospermophilus* (Fig. 7) and *Ammospermophilus* represent multisyllabic signals (Fig. 8)."

Again, I suggest re-writing to: "Calls of North American species of the genus *Urocitellus* (Fig. 6) are also tonal calls: monosyllabic (A, B, C), two-syllabic (D), and multisyllabic (E, F). Similarly, the alarm calls of the genera *Otospermophilus* (Fig. 7) and *Ammospermophilus* (Fig. 8) are the multisyllabic calls."

Line 217 add "alarms" after three-syllable

Line 218 add "alarm" before "calls"

Discussion

Line 233

Delete "representing a whistle sound,"

Line 235-236 which has no analogue in any other Eurasian ground squirrel species
It is incorrect, they are present in the Arctic ground squirrel living on Chukotka.

Line 238-240 We analysed the data based on published results, spectrograms and sounds to expand on the previous comparison of alarms produced by Eurasian and North American ground squirrels conducted by Matrosova et al. [8].

So is unclear why the authors did not include in analysis the listed by Matrosova et al. 2012 species of ground squirrels, for which literature data are available.

Line 241

Delete "(whistles)"

Lines 250-251 "The wideband alarm, which we described in the long-tailed ground squirrel, represents a singly produced signal."

I do not understand the sense of this sentence.

Lines 251-254. Such a call category of wideband alarms was not found either in our comparison or in the study of Matrosova et al. [8]. We suggest here that the long-tailed ground squirrel produces a wideband alarm signal which is something unique among Eurasian and North American species.

It is incorrect. Matrosova et al. 2012 (Table 5) indicates that wideband alarm signal was described for the Arctic ground squirrel (Melchior, 1971; Nikolskii, 1984) and for the Californian ground squirrel (Owings, Virginia, 1978).

Line 253

Delete “signal” and “unique”

Line 256

Change “collected in” to “created”

Line 270 [see 5],

This reference is incorrect. Please replace it with Melchior, 1971; Owings, Virginia, 1978 (see Table 1 in this paper); Nikolskii, 1984

Line 276

Change “signals” with “calls”

Lines 281-282 “Tonal alarms could be assigned correctly to the other species of the genus *Urocitellus* (*U. beldingi*) with 70% success [25].”

I do not understand the sense of this sentence, please re-write.

Lines 289-290. a wideband alarm, which has no similar analogue among ground squirrels either from Eurasia or North America.

It is incorrect; it was described for the Arctic ground squirrel.

Lines 290-291. “Such findings indicate a phylogenetic signal present across alarms of ground squirrels when long-tailed squirrels is phylogenetically”

Sense is unclear, please re-write and correct the typo of ground

Lines 384-386 Delete

Line 397. Scatterplot of tonal alarms (whistles) from 13 individuals.

Please change to Scatterplot on the basis of two first discriminant functions of DFA for whistle alarms from 13 individuals

Line 399. Scatterplot of wideband alarms from 25 individuals.

Please change to Scatterplot on the basis of two first discriminant functions of DFA for wideband alarms from 25 individuals. Please provide a legend with ID number of animals on Fig 4, similarly to Fig 3.

Additional file 1. *Spermophilus suslicus*

This species produces only the monosyllabic whistle alarms. The authors indicate the minimum frequency large than the maximum frequency. Reference to Schneiderova & Policht 2012 *Naturwissenschaften* is incorrect, it is on different species.

Review form: Reviewer 2

Is the manuscript scientifically sound in its present form?

No

Are the interpretations and conclusions justified by the results?

No

Is the language acceptable?

Yes

Do you have any ethical concerns with this paper?

No

Have you any concerns about statistical analyses in this paper?

Yes

Recommendation?

Major revision is needed (please make suggestions in comments)

Comments to the Author(s)

General comments:

This is an interesting manuscript touches on important problem about the role of individuality in the alarm calls of mammals. Ground-dwelling sciurids are a traditional model group for zoological research in various aspects.

The strongest aspect of the study, in my opinion, is the assessment of intraspecific individual differences, both in tonal and broadband calls. It is this part of the work that should be emphasized and developed first. Early studies were carried out at the species level and did not account for individual differences.

However, there were places where the article was problematic and unclear. Below are the areas which I found problematic.

1. The main drawback of the work I see is the superficial study of the information available in the literature. The authors present as the main result of the work the presence of two types of alarm calls in the long-tailed ground squirrel, however, this fact has long been well described. The first descriptions were made by Alexander Nikol'skii (1979, 1984). Also, not all Eurasian species for which there are publications (*S. pygmeus*, *S. mayor*, *S. erythrogegens*, *U. parryi*) were taken for interspecies comparison. The absence of *U. parryi* greatly weakens the article, since the Arctic ground squirrel also live in Asia and belong to long-tailed ground squirrels.

2. The results of the study of individual traits require the use of an equal or similar number of sounds from each animal, the authors do not indicate this important information. The DFA should include an equal number of calls from an individual to obtain reliable results. It is necessary to write in the methods clearly how many individuals produced both types of alarm calls, from how many - only tonal, how many - only wideband. If each animal produced only one type of alarm call (either tonal or wideband), this should also be written clearly.

It is also advisable to add information about the number of examined individuals and calls to the Abstract section.

3. Phylogenetic approach to the species pattern of the cry of alarm is an interesting but methodologically difficult task. The different structure of two types of alarm calls makes comparison difficult. Combining tonal and wideband alarm calls can only be correct when using completely identical call parameters (such as note duration). Comparison of the frequency parameters selected by the authors is incorrect, since in wideband alarm calls one can speak of dominant frequencies, the maximum and minimum fundamental frequencies cannot be measured. In addition, it is not clear how to correctly compare with different numbers of notes. Thus, the last section of the results and the dendrogram of call similarity needs to be seriously revised or removed.

Minor remarks

Line 73. Are all the individuals examined adults? What is the likelihood of sampling big juveniles in July since the animals were unmarked?

Line 108-109, line 132, lines 137-138. Please provide a clear and correct description of the parameters used. Describe separately what was measured and separately what was calculated

from the measurements (e.g. Frequency Modulation parameters) or how the primary measurements were interpreted (such as Min Frequency and Max Frequency).

The number of parameters for wideband calls looks excessive. I advise you to remove uninformative parameters from the article.

Lines 130-131 «The remaining 16 variables were entered into Discrimination Function Analysis». Enter the table with the DFA data and the remaining parameters. You can add this data to Supplementary.

Lines 108-115 Give the number of animals and the number of whistles here. How much from each individual?

Line 135-142. It is necessary to describe in detail the method of sound processing for interspecies comparison. Here you should describe where the alarm calls came from, how much from each species, how the parameters were measured. How were the durations in alarm cries with different numbers of elements counted? It is impossible to measure the parameters of the fundamental frequency for wideband calls, therefore, it is incorrect to combine them with tonal calls and probably should be considered separately or excluded from the scheme.

I highly recommend introducing the table «Additional_file_1._A» into the main body of the article for a better understanding of the material. In this case, the names of acoustic parameters in the table should be brought in line with the text of the article. You should also enter in this Table the number of individuals analyzed for each species.

line 167 Give the spread in the number of studied sounds from each individual (average, from-to).

Line 182. Please, write the number of sounds from each individual and enter in the appropriate place in the Methods how many individuals were excluded from the analysis (probably 31-25 = 6 individuals).

Lines 212-213. Information about the source of the screams should be moved to the Methods section.

Lines 235-236. The Eurasian species of ground squirrels have wideband calls similar in structure in their vocal repertoire, but they are not used as an alarm calls.

Lines 266-268. The position of *S.suslicus* and *S.fulvus* on the dendrogram completely contradicts the phylogenetic reconstruction based on mtDNA. In addition, the alarm calls of *S.suslicus* are monosyllabic, and in the diagram it is grouped with multisyllabic species.

On the whole, speculation about the phylogenetic signal of the alarm calls looks speculative.

Figures

1. On all spectrograms there is no “Y” axis signature - sign “Frequency (kHz)”.
2. For a good perception of the article material, I would like to see different types of alarm calls nearby. I advise you to reduce the number of illustrations by grouping them into complex figures.
3. In Figure 2, label the individual numbers consistently with Figure 1. You can combine them into one figure with two parts (A, B).
4. Figure 3-4 is reasonable to combine into one.
5. Figure 9. Normal font and italic are important information here, but they are very hard to see. Why is there no *S. erythrognys* in Figure 9? Why is RIC (*Urocitellus richardsonii*) shown twice in the figure?

Tables

Table 2 refers to the methods, and is referred to in the text earlier than Table 1, which contains the results. Change places and table numbers.

Despite my criticism, I believe that this work after processing will be very useful for further research in the field of animal science, as it provides new reliable field research data.

Decision letter (RSOS-200147.R0)

Dear Ms Hambálková

The Editors assigned to your paper RSOS-200147 "Individual variation in squirrel alarm: a unique signal among Eurasian and American ground squirrels" have now received comments from reviewers and would like you to revise the paper in accordance with the reviewer comments and any comments from the Editors. Please note this decision does not guarantee eventual acceptance.

Please submit your revised manuscript and required files (see below) no later than 21 days from today's (ie 29-Sep-2020) date. Note: the ScholarOne system will 'lock' if submission of the revision is attempted 21 or more days after the deadline. If you do not think you will be able to meet this deadline please contact the editorial office immediately.

on behalf of Professor Len Thomas (Associate Editor) and Pete Smith (Subject Editor)
openscience@royalsociety.org

Associate Editor Comments to Author (Professor Len Thomas):

Comments to the Author:

Thank-you for your submission to RSOS. We have received two informed reviews of your manuscript and both reviewers agree that your work is very suitable for publication, but make substantive comments and suggestions for improvement. I am therefore recommending acceptance conditional on major revisions to account for these comments. In particular, both reviewers note that the literature review in the paper is too narrow and the methods are insufficiently documented. Regarding the latter point, please note in making your revisions that there is effectively an unlimited space for supplemental materials such as spectrograms, etc. So, please ensure in your revision that the acoustic methods and results are fully documented, making good use of the facility to attach supplemental materials to the paper. Please also take account of the other reviewer comments, such as the importance of individuality from both reviewers and the comments on phylogenetic approach to species pattern from reviewer 2.

Together with any resubmission, please include a letter describing point-by-point how you have dealt with reviewer comments. If this and the revision are satisfactory, I may not need to send the manuscript out for another round of review. I look forward to seeing your revised manuscript.

Reviewer comments to Author:

Reviewer: 1

Comments to the Author(s)

In this study, the authors investigate the acoustics and individuality of two types of the alarm call in the Eurasian ground squirrel species *Urocitellus undulatus*, closely related phylogenetically to North American ground squirrels. The authors discuss the similarities and differences in the acoustics of the alarm call and compare the individuality of the two call types. In addition, the authors compare the alarm calls of *Urocitellus undulatus* with the alarm calls of other species of ground squirrels of Eurasia and North America. This study is very interesting for the broad scientific readership working with diversity of alarm calls across taxa.

Overall, the MS has a high scientific value. However, it is not well-written. The collected by the authors excellent material deserves a more accurate writing.

I have a few main concerns to this MS.

First concern. This study advertise the comparative analysis of alarm calls across species.

However, the authors did not include a large body of literature, describing the alarm calls of many species of Eurasian and North American ground squirrels. This ignored body of literature includes the papers, describing in the first time the wideband alarm calls of the long-tailed and the Arctic ground squirrels and tracking the bioacoustical relationships between the Eurasian and North American species, what is the main focus of the MS. These studies should be obligatory considered in the MS. References to these study are provided in the referee comments below.

Second concern. Methods are non-transparent. Remains unclear, how the animals were recorded, how the caller age class was determined, how the calls were selected for analysis and how the acoustic variables of the calls were measured. The authors should obligatory provide the large-scale illustrative spectrograms showing which acoustic variables were measured for each call type. Descriptions of the acoustic variables, used for inter-species comparison of the alarm calls of ground squirrels, are lacking. Remains unclear, from where and how the data introduced by the authors in the cluster-analysis, were obtained, what leads to great doubts in the obtained results.

Third concern is related to the acoustic analysis. The provided by the authors description of the measured acoustic variables does not allow to compare them with those commonly measured in the similar papers. The filtering mode, applied by the authors to the acoustic file, should affect the acoustic structure of both the whistle and the wideband alarm calls. Judging by Figure 2, this procedure just will cut off the lower halves of the wideband alarms. The authors should explain this and probably re-calculate their results without applying this damaging filtering.

Fourth concern is related to statistic analyses. I suggest calculating the average values for each acoustic variable for each individual and then use them for the descriptive statistics. Such sample is more balanced than data pooling, used in the MS. The results of DFA are dependent from the sample size. Probably, that the low value of correct assignment for some individuals is related to the misbalanced call samples. I also found that calculation of DFA classification by chance was made incorrectly and should be re-calculated.

I suggest to change the title, as it does not reflect the content of the MS. I also suggest to add some short information about the results of analyses of individuality in Abstract. The language is overall acceptable (for the exclusion of a few unclear sentences) and was corrected by the professional proofreading with editing services. The revised MS should be then re-reviewed again.

Ethics

Line 22 Please correct typos in the word "regional"

Title

The title is not perfectly clear, what the authors mean under the unique signal? Do you indeed compare individual identity of the alarm call between American and Eurasian squirrels? Please correct the title. Probable variant: "Individual-based acoustic variation of the alarm call in the long-tailed ground squirrel suggests a closer relationship with North American than Eurasian ground squirrels".

Abstract

Abstract says nothing regarding the individual variation, please add you results about individuality in Abstract.

Line 17 Delete "the" before "vocalization"

Line 19 add "ground squirrel" after "species"

Lines 20-21. We found a unique signal, which confirms the closer relationship of Eurasian ground squirrels to North American species, rather than other species from Eurasia. This finding is in concordance with previous molecular results.

If I understood correctly the sense of this sentence, it should be corrected as follows "We found that the alarm call of the long-tailed ground squirrel was distinctive among Eurasian species of ground squirrels but was similar with those of North American species of ground squirrels" Please re-write.

Lines 22-24 The split of Eurasian ground squirrels into two genera - Spermophilus and Urocitellus - based on genetics is supported by vocalization.

I suggest re-writing to "We discuss that these acoustic data confirm the genetic-based separation of Eurasian ground squirrels to the two genera, Spermophilus and Urocitellus."

Introduction

Lines 30-31. Vocalization in both Eurasian and North American ground squirrels has been intensively studied, especially during the last decade.
References are necessary at the end of this sentence.

Line 37. The long-tailed ground squirrel species

Unclear, what the authors mean here, one species - the long-tailed ground squirrel of both species of genus Urocitellus (the long-tailed and the Arctic)

Line 45. Such a situation is unique among ground squirrels studied so far.

Delete this sentence

Lines 46-47 The question at hand is whether alarm call in long-tailed squirrel could be a result of phylogenetic correlation with other species.

This sentence is unclear. Do you mean the result of phylogenetic relationship with other species?

Lines 51-52 Correct citations. European - delete 16 and insert 17, Anatolian - change 12 to 13

Lines 49-53. In reality, descriptions of the alarm calls of *Spermophilus* ground squirrels are available for more number of species. As this study pretends to provide a reviewing comparison, the authors should obligatorily include these papers in the MS. Most of these papers are in Russian, however, two of the authors are native Russian speakers, so the reading these papers will not be difficult for them. Most of these studies are rather old and the acoustic analyses are weak, but they cannot be ignored. References are provided below.

Nikolskii, 1979. Species specificity of alarm call in sousliks of (*Citellus*, *Sciuridae*) Eurasia. *Zool. Zhurnal*, v. 58, N 8, p. 1183-1194 and

Nikolskii, 1984. Acoustic signals of mammals in the evolutionary process. Moscow, Nauka. Description of species-specific alarm calls of ground squirrels of Eurasia *S. citellus*, *S. suslicus*, *S. xanthoprimum*, *S. pygmaeus*, *S. alaschanicus*, *S. erythrognys*, *S. maior*, *S. fulvus*, *S. relictus*. In addition, spectrograms of the alarm calls of *U. undulatus* and *U. parryi* are provided. A description of the bioacoustical relationships of the two last species with ground squirrels of North America.

Nikolskii et al., 1984. The alarm call in F1 hybrids *Citellus pygmaeus* X *C. suslica* (*Sciuridae*, *Rodentia*). *Zool Zhurnal*, v. 63, N 8, p. 1216-1225.

Nikolskii, Starikov, 1997. Variability of alarm call in *Spermophilus maior* and *Spermophilus erythrognys* (*Rodentia*, *Sciuridae*) within contact zone in Kurgan district. *Zool Zhurnal*, v. 76, N 7, p. 845-857.

Nikolskii et al., 2007. Geographical variability of the little ground squirrel (*Spermophilus pygmaeus*): a bioacoustic analysis. *Zool Zhurnal*, v. 86, N 11, p. 1379-1388.

Nikol'skii, 2019. Sex accent and biphonation in the sound signal of ground squirrels (*Mammalia*, *Rodentia*). *Doklady Biological Sciences*, 2019, V. 487, p. 119-123. Russian Text in *Doklady Akademii Nauk*, 2019, V. 487, No. 6.

Line 55 Delete "especially". Otherwise, it sounds as that individual-level variation is higher than the species-level variation, what is not the case in ground squirrels.

Line 56. Correct citations. Add 6 and 9

Line 57-58. a whistle sound with simple tonal structure

Please delete "whistle" here. Whereas calls of some rodents indeed represent the aerodynamic whistles by the production mechanism (Riede, 2011, 2013; Pasch et al. 2017; Riede et al. 2017; Riede, Pasch 2020) and in ground squirrels probably too, this production mode has not yet been confirmed for the alarm calls of ground squirrels.

Please re-write this sentence as "tonal calls of very simple structure"

Line 66 "to test the individual variation in alarm calls"

Individual variation was not mentioned in Abstract and appeared in the first time only in the aim. Please add something you findings regarding the individuality of alarm calls in Abstract.

Line 67 strongly species-specific. Please add reference to Nikolskii 1979, 1984

Line 67 supposed to be innate. Please add reference to Matocha, 1975. Vocal communication in ground squirrels, genus *Spermophilus*. PhD thesis, Graduate Faculty of Texas Tech University.

Methods

Line 82 the first study site. Change to Sarma village

Line 91-94. Please indicate the age of animal callers (adults or not) and how they were aged. Please indicate the distance during recording (from - to). Please indicate the approximate duration of recording. Please indicate what the observer did during the recording, because the behaviour of researcher affect animal vocal activity and producing either the whistle or the wideband alarms. Please indicate, what calls the animals produced: only the whistle alarms, only the wideband alarms or both call types. Please indicate, how many animals you recorded and how they were distributed by the 14 colonies.

Most important - please indicate, by which traits you determine that you record the focal animal rather than the hidden caller. My experience of recording ground squirrels in the field suggests that mistake is highly likely, especially if the silent focal animal is partially hidden or is moving, and the second (caller) is nearby but hidden perfectly from the researcher (by grass, bush etc.). This is critically important for the results on individuality of the alarm calls.

Lines 95-96 "The Animal Care and Use Committee of the Czech Ministry of the Environment approved this research."

Please transfer this sentence in the Ethics section.

Line 102 "Whistles"

Change "Whistles" to "Whistle alarms" throughout. The authors should indicate here that long-tailed ground squirrels produce two types of alarm calls, the whistle alarm and the wideband alarm, and refer to the respective figures with spectrograms.

Line 103 We analysed whistles of the highest quality.

Please provide the number of selected whistle alarms (as you did below for the wideband alarms).

Please provide the criteria of the highest quality (for example, calls with high signal-to noise ratio, not broken with wind, not overlapped with alien calls or noises etc.).

Line 103. Please indicate in detail, how many whistle alarms were selected for analyses. From how many individuals. How many calls per animal (Mean, SD). How calls were selected within individual (from one of from a few series, following each other or not).

Lines 104 and further. software

Please provide the firm, headquarters city and country of producer, this is a common claim of all journals.

Lines 106-107 Background noise was filtered out using both high-pass (1.3–2.9 kHz) and low-pass filters (5.2–7.7 kHz).

The authors should write here, why these frequency ranges were filtered out. Remains unclear, whether such filtering mode affected the important parts of the studied calls. From Table 1 follows, that maximum values of the start fundamental frequency (6.65 kHz) are within the zone which was filtered out.

Lines 107-115

This list of measured acoustic parameters is non-informative. Unclear what the authors indeed measured. What do you mean under "frequency"? The fundamental frequency? Or the automatically measured peak frequency (frequency of maximum amplitude)? The authors should obligatory provide the figure illustrating all measured acoustic variables.

Line 118 "We selected..."

Please provide criteria for this selection of wideband alarm calls for analysis. Please indicate in detail, how many wideband alarms was selected per animal (Mean, SD). How calls were selected within individual (from one or a few call series, following each other or not).

Lines 119-120 Why 1.6 kHz high-pass filtering was applied for filtration? What was frequency range of the calls? Were meaningful call part damaged due to this filtering? Judging by spectrograms presented on Figure 2, the authors cut-off the lower halves of calls and then only analyses the upper halves. This is very strange and demands a detailed explanation. Moreover, many values of acoustic variables indicated in Table2 are lower 1600 Hz. How it is possible?

Line 125 Reference to Table 2 appears in text earlier then the reference to Table 1. Please correct this.

Line 124 We measured 22 parameters

How these 22 parameters were measured? Manually? Automatically?

Lines 124-125, Table 2. This description of measured parameters in insufficient. Please provide a detailed description of parameters being measured and provide a figure with illustrative spectrogram showing the measured acoustic variables. The used acoustic variables are unclear and are incompatible with the acoustic variables used in other papers on the alarm calls of ground squirrels, what makes the results of MS incomparable with other published studies. Please, make the descriptions of the acoustic variables clear at least for reviewer - bioacoustic researcher.

For example, 1st Quartile Frequency – this is a part of call spectrum, covering 25% of the total call energy. However, Table 2 describes the 1st Quartile Frequency as "The frequency that divides the selection into two frequency intervals containing 25% and 75% of the energy in the signal". How Q1F differs from Q1FRel, which is determined as "The frequency that divides the selection into two frequency Intervals containing 25% and 75% of the energy in the signal relative to the frequency range of the signal"?

"Maximum entropy calculated from each frame". What is frame and how many frames?

"The difference between the 5% and 95% frequencies". Bandwidth is the width of the frequency band of call spectrum. What you understand under "frequency"? How 1.6 kHz high-pass filtering affects the bandwidth?

Center Frequency "The frequency that divides the selection into two frequency intervals of equal energy". What frequency, fundamental or peak? Or this is the median of call power spectrum?

.... etc.

Line 128. Individuals with a lower number of calls ($n < 7$)

What calls, whistle alarms or wideband alarms?

Line 128-134. The results of a DFA are dependent from the sample size. I would suggest to balance the data set by number of calls per individual, included in each DFA.

Line 137-138. Six acoustic variables (Call duration, Element duration, Start Frequency, End Frequency, Min Frequency and Max Frequency)

It is necessary to provide a detailed description of these acoustic variables. Call duration – was this the duration of a series of repetitive calls? Frequency – was this the fundamental or the peak frequency? How it was measured when the alarm call represented a series of repetitive calls? Without clear methods, results cannot be understood.

Results

Line 146 and further. Alarm call description

As you analysed evidently different call numbers from different individuals, would be better to calculate the average values of acoustic variable for each individual and then use them in the descriptive statistics in Table 1. Such sample will be more balanced for calculating the averages per species than data pooling, used in MS.

Line 147 tonal whistles

Change to whistle alarms throughout, uniformly with wideband alarms.

Lines 148-149 "calls with a tonal frequency modulated structure formed of one syllable only"
I suggest re-writing to "one-syllable calls with a visible modulated fundamental frequency"

Line 149 mean \pm SD

Transfer this in Statistics section, e.g. "All means were indicated as mean \pm SD".

Line 156 the second type of alarm signal

Please avoid synonyms, moreover "signal" is incorrect term here. I suggest rewriting to "the second alarm call type,"

Line 166 Tonal alarms (Whistles)

Do not use synonyms, change with whistle alarms

Lines 167-168. We excluded the highly correlated 2nd Frequency from all subsequent analysis. In Methods (Lines 131-132) you wrote that all 6 acoustic variables of whistle alarms were included in DFA. Please correct.

Lines 167-179. The discrimination analyses considered 269 whistles from 13 individuals. The results of a DFA are dependent from the sample size. Please provide a table with DFA results, including the number of whistle alarms from each of the 13 individuals and percentages of correct assignment of calls to each individual. Probably, the low value of correct assignment for some individuals is related to misbalanced sample.

Lines 178-179. Such classification results were much higher in comparison to classification by chance (7.7%).

Classification by chance was calculated incorrectly. Please use either the method by Solow, 1990 (Ecology, 71:2379-2382), or by Mundry, Sommer, 2007 (Animal Behaviour, 74:965-976)

Lines 182-183. The model contained ten acoustic variables.

Again, earlier in Methods (Line 130) is indicated that 16 acoustic variables was included in analysis. Please correct.

Lines 182-192. We entered 591 wideband alarms from 25 individuals into the following analysis. The results of a DFA are dependent from the sample size. Please provide a table with DFA results, providing the numbers of wideband alarms from each of the 25 individuals and percentages of correct assignment to each individual. Probably, the low value of correct assignment for some individuals is related to misbalanced sample.

Lines 191-192. These results were much higher than classification by chance (4%).

Classification by chance was calculated incorrectly. Please use either the method by Solow, 1990 (Ecology, 71:2379-2382), or by Mundry, Sommer, 2007 (Animal Behaviour, 74:965-976)

Line 194. Fig. 4

Reference to Fig 3 and description of this figure are lacking in the MS.

Line 200 and further. Species comparison

Unfortunately, this interesting section is written based on incomplete and unclear how obtained data. Accordingly, the results of this section have a negligible meaning. This section should be re-worked and re-calculated perfectly.

First, the authors should provide in Methods a detailed description of the used acoustic variables. For example, what frequency was measured (fundamental or peak?) in the wideband alarms of the long-tailed ground squirrel and how it was measured? And how this measurement agrees with similar measurements in the tonal alarms of calls of other species. I think that the results of Fig 9 about the separate position of the wideband alarm of the long-tailed ground squirrel directly result from the incorrect measurements of "frequency".

Second, remains unclear, from where and how the data introduced by the authors in cluster-analysis were obtained. Were they taken from the papers? Or from the own measurements of calls taken from web-sites? Of how many calls and from how many individuals?

Third, interspecies comparisons of the alarm calls of ground squirrels (Fig. 9 and Additional file 1) are evidently incomplete. Alarms of a few species available from literature (and even presented on Figs 5-8, but not included in Fig 9) should be included. Data of the cue papers on the Arctic ground squirrel are lacking, although they are critically important for comparison of Eurasian and North American ground squirrels (Melchior, 1971, *Oecologia*, 7:184-190; Nikolskii, 1979, 1984). The Arctic ground squirrel lives in tundra on both sides of Bering Strait and produces both the whistle and wideband alarms, very similar with the respective call types of the long-tailed ground squirrel. Without inclusion of these data, the picture of uniqueness of alarm calls of long-tailed ground squirrel is distorted.

The missing in MS references on the alarm calls of ground squirrels are available in Matrosova et al. 2012 (cited in MS) and in a new review on vocalizations of ground squirrels (Newar, Bowman, 2020, *Front. Ecol. Evol.*, 2020, v. 8: 00193. doi: 10.3389/fevo.2020.00193), especially in Supplementary to this review.

Lines 203-205 These calls represent tonal calls: monosyllabic whistles (A, D, E), two-syllable whistles (B, C), and multisyllabic chirps (F) and churrs (G).

Whistles, chirps, churrs: this terminology is redundant and only confuses the reader. I suggest re-writing to: "These alarm calls are all tonal calls: monosyllabic (A, D, E), two-syllabic (B, C), and multisyllabic (F, G)."

Lines 205-208 "Calls of North American species (Fig. 6) also represent monosyllabic whistles (A, B, C), two-syllable whistles (D), and multisyllabic chirps (E, F). Similarly, calls of the genera *Otospermophilus* (Fig. 7) and *Ammospermophilus* represent multisyllabic signals (Fig. 8)."

Again, I suggest re-writing to: "Calls of North American species of the genus *Urocitellus* (Fig. 6) are also tonal calls: monosyllabic (A, B, C), two-syllabic (D), and multisyllabic (E, F). Similarly, the alarm calls of the genera *Otospermophilus* (Fig. 7) and *Ammospermophilus* (Fig. 8) are the multisyllabic calls."

Line 217 add "alarms" after three-syllable

Line 218 add "alarm" before "calls"

Discussion

Line 233

Delete "representing a whistle sound,"

Line 235-236 which has no analogue in any other Eurasian ground squirrel species
It is incorrect, they are present in the Arctic ground squirrel living on Chukotka.

Line 238-240 We analysed the data based on published results, spectrograms and sounds to expand on the previous comparison of alarms produced by Eurasian and North American ground squirrels conducted by Matrosova et al. [8].

So is unclear why the authors did not include in analysis the listed by Matrosova et al. 2012 species of ground squirrels, for which literature data are available.

Line 241

Delete "(whistles)"

Lines 250-251 "The wideband alarm, which we described in the long-tailed ground squirrel, represents a singly produced signal."

I do not understand the sense of this sentence.

Lines 251-254. Such a call category of wideband alarms was not found either in our comparison or in the study of Matrosova et al. [8]. We suggest here that the long-tailed ground squirrel produces a wideband alarm signal which is something unique among Eurasian and North American species.

It is incorrect. Matrosova et al. 2012 (Table 5) indicates that wideband alarm signal was described for the Arctic ground squirrel (Melchior, 1971; Nikolskii, 1984) and for the Californian ground squirrel (Owings, Virginia, 1978).

Line 253

Delete "signal" and "unique"

Line 256

Change "collected in" to "created"

Line 270 [see 5],

This reference is incorrect. Please replace it with Melchior, 1971; Owings, Virginia, 1978 (see Table 1 in this paper); Nikolskii, 1984

Line 276

Change "signals" with "calls"

Lines 281-282 "Tonal alarms could be assigned correctly to the other species of the genus *Urocitellus* (*U. beldingi*) with 70% success [25]."

I do not understand the sense of this sentence, please re-write.

Lines 289-290. a wideband alarm, which has no similar analogue among ground squirrels either from Eurasia or North America.

It is incorrect; it was described for the Arctic ground squirrel.

Lines 290-291. "Such findings indicate a phylogenetic signal present across alarms of ground squirrels when long-tailed squirrels is phylogenetically"

Sense is unclear, please re-write and correct the typo of ground

Lines 384-386 Delete

Line 397. Scatterplot of tonal alarms (whistles) from 13 individuals.

Please change to Scatterplot on the basis of two first discriminant functions of DFA for whistle alarms from 13 individuals

Line 399. Scatterplot of wideband alarms from 25 individuals.

Please change to Scatterplot on the basis of two first discriminant functions of DFA for wideband alarms from 25 individuals. Please provide a legend with ID number of animals on Fig 4, similarly to Fig 3.

Additional file 1. *Spermophilus suslicus*

This species produces only the monosyllabic whistle alarms. The authors indicate the minimum frequency large than the maximum frequency. Reference to Schneiderova & Policht 2012 Naturwissenschaften is incorrect, it is on different species.

Reviewer: 2

Comments to the Author(s)

General comments:

This is an interesting manuscript touches on important problem about the role of individuality in the alarm calls of mammals. Ground-dwelling sciurids are a traditional model group for zoological research in various aspects.

The strongest aspect of the study, in my opinion, is the assessment of intraspecific individual differences, both in tonal and broadband calls. It is this part of the work that should be emphasized and developed first. Early studies were carried out at the species level and did not account for individual differences.

However, there were places where the article was problematic and unclear. Below are the areas which I found problematic.

1. The main drawback of the work I see is the superficial study of the information available in the literature. The authors present as the main result of the work the presence of two types of alarm calls in the long-tailed ground squirrel, however, this fact has long been well described. The first descriptions were made by Alexander Nikol'skii (1979, 1984). Also, not all Eurasian species for which there are publications (*S. pygmeus*, *S. mayor*, *S. erythrognys*, *U. parryi*) were taken for interspecies comparison. The absence of *U. parryi* greatly weakens the article, since the Arctic ground squirrel also live in Asia and belong to long-tailed ground squirrels.

2. The results of the study of individual traits require the use of an equal or similar number of sounds from each animal, the authors do not indicate this important information. The DFA should include an equal number of calls from an individual to obtain reliable results. It is necessary to write in the methods clearly how many individuals produced both types of alarm calls, from how many - only tonal, how many - only wideband. If each animal produced only one type of alarm call (either tonal or wideband), this should also be written clearly.

It is also advisable to add information about the number of examined individuals and calls to the Abstract section.

3. Phylogenetic approach to the species pattern of the cry of alarm is an interesting but methodologically difficult task. The different structure of two types of alarm calls makes comparison difficult. Combining tonal and wideband alarm calls can only be correct when using completely identical call parameters (such as note duration). Comparison of the frequency parameters selected by the authors is incorrect, since in wideband alarm calls one can speak of dominant frequencies, the maximum and minimum fundamental frequencies cannot be measured. In addition, it is not clear how to correctly compare with different numbers of notes. Thus, the last section of the results and the dendrogram of call similarity needs to be seriously revised or removed.

Minor remarks

Line 73. Are all the individuals examined adults? What is the likelihood of sampling big juveniles in July since the animals were unmarked?

Line 108-109, line 132, lines 137-138. Please provide a clear and correct description of the parameters used. Describe separately what was measured and separately what was calculated from the measurements (e.g. Frequency Modulation parameters) or how the primary measurements were interpreted (such as Min Frequency and Max Frequency).

The number of parameters for wideband calls looks excessive. I advise you to remove uninformative parameters from the article.

Lines 130-131 «The remaining 16 variables were entered into Discrimination Function Analysis». Enter the table with the DFA data and the remaining parameters. You can add this data to Supplementary.

Lines 108-115 Give the number of animals and the number of whistles here. How much from each individual?

Line 135-142. It is necessary to describe in detail the method of sound processing for interspecies comparison. Here you should describe where the alarm calls came from, how much from each species, how the parameters were measured. How were the durations in alarm cries with different numbers of elements counted? It is impossible to measure the parameters of the fundamental frequency for wideband calls, therefore, it is incorrect to combine them with tonal calls and probably should be considered separately or excluded from the scheme.

I highly recommend introducing the table «Additional_file_1._A» into the main body of the article for a better understanding of the material. In this case, the names of acoustic parameters in the table should be brought in line with the text of the article. You should also enter in this Table the number of individuals analyzed for each species.

line 167 Give the spread in the number of studied sounds from each individual (average, from-to).

Line 182. Please, write the number of sounds from each individual and enter in the appropriate place in the Methods how many individuals were excluded from the analysis (probably 31-25 = 6 individuals).

Lines 212-213. Information about the source of the screams should be moved to the Methods section.

Lines 235-236. The Eurasian species of ground squirrels have wideband calls similar in structure in their vocal repertoire, but they are not used as an alarm calls.

Lines 266-268. The position of *S.suslicus* and *S.fulvus* on the dendrogram completely contradicts the phylogenetic reconstruction based on mtDNA. In addition, the alarm calls of *S.suslicus* are monosyllabic, and in the diagram it is grouped with multisyllabic species.

On the whole, speculation about the phylogenetic signal of the alarm calls looks speculative.

Figures

1. On all spectrograms there is no “Y” axis signature - sign “Frequency (kHz)”.
2. For a good perception of the article material, I would like to see different types of alarm calls nearby. I advise you to reduce the number of illustrations by grouping them into complex figures.
3. In Figure 2, label the individual numbers consistently with Figure 1. You can combine them into one figure with two parts (A, B).
4. Figure 3-4 is reasonable to combine into one.
5. Figure 9. Normal font and italic are important information here, but they are very hard to see. Why is there no *S. erythrognys* in Figure 9? Why is RIC (*Urocitellus richardsonii*) shown twice in the figure?

Tables

Table 2 refers to the methods, and is referred to in the text earlier than Table 1, which contains the results. Change places and table numbers.

Despite my criticism, I believe that this work after processing will be very useful for further research in the field of animal science, as it provides new reliable field research data.

===PREPARING YOUR MANUSCRIPT===

===PREPARING YOUR REVISION IN SCHOLARONE===

- 1) One version identifying all the changes that have been made (for instance, in coloured highlight, in bold text, or tracked changes);
 - 2) A 'clean' version of the new manuscript that incorporates the changes made, but does not highlight them.
 - An individual file of each figure (EPS or print-quality PDF preferred [either format should be produced directly from original creation package], or original software format).
 - An editable file of each table (.doc, .docx, .xls, .xlsx, or .csv).
 - An editable file of all figure and table captions.
- Note: you may upload the figure, table, and caption files in a single Zip folder.
- Any electronic supplementary material (ESM).
 - If you are requesting a discretionary waiver for the article processing charge, the waiver form must be included at this step.
 - If you are providing image files for potential cover images, please upload these at this step, and inform the editorial office you have done so. You must hold the copyright to any image provided.
 - A copy of your point-by-point response to referees and Editors. This will expedite the preparation of your proof.

- Ensure that your data access statement meets the requirements at <https://royalsociety.org/journals/authors/author-guidelines/#data>. You should ensure that you cite the dataset in your reference list. If you have deposited data etc in the Dryad repository, please include both the 'For publication' link and 'For review' link at this stage.
- If you are requesting an article processing charge waiver, you must select the relevant waiver option (if requesting a discretionary waiver, the form should have been uploaded at Step 3 'File upload' above).
- If you have uploaded ESM files, please ensure you follow the guidance at <https://royalsociety.org/journals/authors/author-guidelines/#supplementary-material> to include a suitable title and informative caption. An example of appropriate titling and captioning may be found at https://figshare.com/articles/Table_S2_from_Is_there_a_trade-off_between_peak_performance_and_performance_breadth_across_temperatures_for_aerobic_sc_ope_in_teleost_fishes_/3843624.

Author's Response to Decision Letter for (RSOS-200147.R0)

See Appendix A.

Decision letter (RSOS-200147.R1)

Dear Ms Hambálková,

It is a pleasure to accept your manuscript entitled "Individual-based acoustic variation of the alarm calls in the long-tailed ground squirrel" in its current form for publication in Royal Society Open Science. The comments of the reviewer(s) who reviewed your manuscript are included at the foot of this letter.

on behalf of Professor Len Thomas (Associate Editor) and Pete Smith (Subject Editor)
openscience@royalsociety.org

Associate Editor Comments to Author (Professor Len Thomas):

Comments to the Author:

Thank-you for doing such a thorough job with your resubmission and detailed cover letter.
Having reviewed your responses, I am happy now to recommend acceptance.

Appendix A

Reviewer comments to Author:

Reviewer: 1

First concern. This study advertise the comparative analysis of alarm calls across species. However, the authors did not include a large body of literature, describing the alarm calls of many species of Eurasian and North American ground squirrels. This ignored body of literature includes the papers, describing in the first time the wideband alarm calls of the long-tailed and the Arctic ground squirrels and tracking the bioacoustical relationships between the Eurasian and North American species, what is the main focus of the MS. These studies should be obligatory considered in the MS. References to these study are provided in the referee comments below.

Second concern. Methods are non-transparent. Remains unclear, how the animals were recorded, how the caller age class was determined, how the calls were selected for analysis and how the acoustic variables of the calls were measured. The authors should obligatory provide the large-scale illustrative spectrograms showing which acoustic variables were measured for each call type.

Descriptions of the acoustic variables, used for inter-species comparison of the alarm calls of ground squirrels, are lacking. Remains unclear, from where and how the data introduced by the authors in the cluster-analysis, were obtained, what leads to great doubts in the obtained results.

Third concern is related to the acoustic analysis. The provided by the authors description of the measured acoustic variables does not allow to compare them with those commonly measured in the similar papers. The filtering mode, applied by the authors to the acoustic file, should affect the acoustic structure of both the whistle and the wideband alarm calls. Judging by Figure 2, this procedure just will cut off the lower halves of the wideband alarms. The authors should explain this and probably re-calculate their results without applying this damaging filtering.

Fourth concern is related to statistic analyses. I suggest calculating the average values for each acoustic variable for each individual and then use them for the descriptive statistics. Such sample is more balanced than data pooling, used in the MS. The results of DFA are dependent from the sample size. Probably, that the low value of correct assignment for some individuals is related to the misbalanced call samples. I also found that calculation of DFA classification by chance was made incorrectly and should be re-calculated.

I suggest to change the title, as it does not reflect the content of the MS. I also suggest to add some short information about the results of analyses of individuality in Abstract. The language is overall acceptable (for the exclusion of a few unclear sentences) and was corrected by the professional proofreading with editing services. The revised MS should be then re-reviewed again.

Ethics

Line 22 Please correct typos in the word "regional"

CORRECTED

Title

The title is not perfectly clear, what the authors mean under the unique signal? Do you indeed compare individual identity of the alarm call between American and Eurasian squirrels? Please correct the title. Probable variant: "Individual-based acoustic variation of the alarm call in the long-tailed ground squirrel suggests a closer relationship with North American than Eurasian ground squirrels".

ACCEPTED. We partly used a suggested title.

Abstract

Abstract says nothing regarding the individual variation, please add you results about individuality in Abstract.

ACCEPTED. We added it into the abstract.

Line 17 Delete “the” before “vocalization”

This part has been deleted.

Line 19 add “ground squirrel” after “species”

This part has been deleted.

Lines 20-21. We found a unique signal, which confirms the closer relationship of Eurasian ground squirrels to North American species, rather than other species from Eurasia. This finding is in concordance with **previous** molecular results.

If I understood correctly the sense of this sentence, it should be corrected as follows “We found that the alarm call of the long-tailed ground squirrel was distinctive among Eurasian species of ground squirrels but was similar with those of North American species of ground squirrels” Please re-write.

This part has been deleted.

Lines 22-24 The split of Eurasian ground squirrels into two genera – Spermophilus and Uroditellus – based on genetics is supported by vocalization.

I suggest re-writing to “We discuss that these acoustic data confirm the genetic-based separation of Eurasian ground squirrels to the two genera, Spermophilus and Uroditellus.”

This part has been deleted.

Introduction

Lines 30-31. Vocalization in both Eurasian and North American ground squirrels has been intensively studied, especially during the last decade.

References are necessary at the end of this sentence.

ACCEPTED. We added the references. (Line 34)

Line 37. The long-tailed ground squirrel species

Unclear, what the authors mean here, one species – the long-tailed ground squirrel of both species of genus Uroditellus (the long-tailed and the Arctic).

ACCEPTED. We added a latin name. (Line 42)

Line 45. Such a situation is unique among ground squirrels studied so far.

Delete this sentence.

ACCEPTED. We deleted the sentence.

Lines 46-47 The question at hand is whether alarm call in long-tailed squirrel could be a result of phylogenetic correlation with other species.

This sentence is unclear. Do you mean the result of phylogenetic relationship with other species?

This part has been deleted.

Lines 51-52 Correct citations. European - delete 16 and insert 17, Anatolian – change 12 to 13

CORRECTED. The order of citations has been adjusted due to the removal or addition of sources.

Lines 49-53. In reality, descriptions of the alarm calls of Spermophilus ground squirrels are available for more number of species. As this study pretends to provide a reviewing comparison, the authors should obligatory include these papers in the MS. Most of these papers are in Russian, however, two of the authors are native Russian speakers, so the reading these papers will not be difficult for them. Most of these studies are rather old and the acoustic analyses are weak, but they cannot be ignored. References are provided below.

ACCEPTED. We added the following part as follows (Lines 54-61):

“...Russet ground squirrel S. major [26,27], Caucasian mountain S. musicus [25], Little ground squirrel S. pygmaeus [25,28], S. alaschanicus [25], Arctic ground squirrel Urocitellus parryii [25], Gray marmot Marmota baibacina [29,30], Steppe marmot Marmota bobak [31]. The basic description of the alarm calls produced by ground squirrels inhabiting the Russian and Asian area included comparative study containing the following species: Urocitellus undulatus, U. parryii, Spermophilus xanthopyrminus, S. musicus, S. pygmaeus, S. alaschanicus, S. suslicus, S. relictus, S. citellus, S. erythrogegens, S. major, S. fulvus and S. rully [30,32].”

Line 55 Delete “especially”. Otherwise, it sounds as that individual-level variation is higher than the species-level variation, what is not the case in ground squirrels.

CORRECTED

Line 56. Correct citations. Add 6 and 9

CORRECTED. The order of citations has been adjusted due to the removal or addition of sources.

Line 57-58. a whistle sound with simple tonal structure

Please delete “whistle” here. Whereas calls of some rodents indeed represent the aerodynamic whistles by the production mechanism (Riede, 2011, 2013; Pasch et al. 2017; Riede et al. 2017; Riede, Pasch 2020) and in ground squirrels probably too, this production mode has not yet been confirmed for the alarm calls of ground squirrels.

Please re-write this sentence as "tonal calls of very simple structure"

CORRECTED. (Line 65)

Line 66 “to test the individual variation in alarm calls”

Individual variation was not mentioned in Abstract and appeared in the first time only in the aim. Please add something you findings regarding the individuality of alarm calls in Abstract.

ACCEPTED. We added DFA results.

Line 67 strongly species-specific. Please add reference to Nikolskii 1979, 1984

ACCEPTED. We added the references.

Line 67 supposed to be innate. Please add reference to Matocha, 1975. Vocal communication in ground squirrels, genus *Spermophilus*. PhD thesis, Graduate Faculty of Texas Tech University.

ACCEPTED. We added the reference.

Methods

Line 82 the first study site. Change to Sarma village

CORRECTED

Line 91-94. Please indicate the age of animal callers (adults or not) and how they were aged.

ACCEPTED. We added the age categories under Methods, Results and Discussion as follows:

Methods (Lines 103-105)

“We categorized two age categories based on the body size: adults and subadults. Subadults were clearly smaller in size, an estimated half the size of adults.”

Results (Line 220-223)

“Age effect

Additionally, we tested a potential distinctness between age and did not find a significant difference in any of acoustic parameter ($0.11 < p < 0.61$; Mann-Whitney U Test) in both call types.”

Discussion (Lines 239-243)

“In our study, both call types did not differ between adults and subadults. Although young individuals differ significantly in size from adults, the fundamental frequency of alarm whistles does not differ in other studied Palearctic ground squirrels [17,35,36]. According to our results, such phenomenon seems to be valid also for wideband alarm signal. Age information could be concealed in alarm calls of Palearctic ground squirrels [36].”

Please indicate the distance during recording (from - to). Please indicate the approximate duration of recording. Please indicate what the observer did during the recording, because the behaviour of researcher affect animal vocal activity and producing either the whistle or the wideband alarms. Please indicate, what calls the animals produced: only the whistle alarms, only the wideband alarms or both call types.

ACCEPTED. We added the information as follows (Lines 103-110):

“We categorized two age categories based on the body size: adults and subadults. Subadults were clearly smaller in size, an estimated half the size of adults. The vocalizations were recorded at distances from 3 to 12 m. Duration of each recording ranged from 3 to 10 minutes, depending on the duration of the calling behavior of the subject. When we met an individual or a group where at least one group member was calling, we stopped and started recording in a sitting position. Each recorded individual used only one type of call, either an alarm whistle or a wideband signal. The use of the alarm type does not appear to be affected by a context that was approximately constant.”

Please indicate, how many animals you recorded and how they were distributed by the 14 colonies.

ACCEPTED. We included this information in the Table S1. Recorded individuals (Supplementary material).

Call type	ID	Age	Locality	Colony	Number of calls
Whistle	1	Subadult	Sarma	3	10
Whistle	2	Adult	Sarma	1	12
Whistle	3	Subadult	Sarma	3	25
Whistle	4	Subadult	Sarma	3	32
Whistle	5	Adult	Sarma	3	20
Whistle	6	Adult	Sarma	7	20
Whistle	7	Adult	Sarma	7	9
Whistle	8	Subadult	Sarma	8	25
Whistle	9	Subadult	Sarma	8	25
Whistle	10	Adult	Sarma	10	16
Whistle	11	Adult	Kurma	14	31
Whistle	12	Adult	Kurma	14	24
Whistle	13	Subadult	Sarma	5	20
Wideband	1	Adult	Sarma	3	21
Wideband	2	Subadult	Sarma	3	28
Wideband	3	Subadult	Sarma	3	33
Wideband	4	Adult	Sarma	3	30
Wideband	5	Adult	Sarma	3	18
Wideband	6	Adult	Sarma	2	31
Wideband	7	Adult	Sarma	1	34
Wideband	8	Subadult	Sarma	1	15
Wideband	9	Adult	Sarma	1	28
Wideband	10	Adult	Sarma	3	25
Wideband	11	Adult	Sarma	7	25
Wideband	12	Adult	Sarma	4	27
Wideband	13	Adult	Sarma	5	25
Wideband	14	Adult	Sarma	5	26
Wideband	15	Adult	Sarma	5	26
Wideband	16	Adult	Sarma	9	25
Wideband	17	Adult	Sarma	12	22
Wideband	18	Subadult	Kurma	13	19
Wideband	19	Adult	Kurma	13	24
Wideband	20	Subadult	Kurma	13	21
Wideband	21	Adult	Kurma	13	23
Wideband	22	Adult	Kurma	13	9
Wideband	23	Adult	Kurma	13	23
Wideband	24	Adult	Kurma	14	12

Table S1. Number of individuals recorded, their age, location of recording, colonies and number of calls.

Most important – please indicate, by which traits you determine that you record the focal animal rather than the hidden caller. My experience of recording ground squirrels in the field suggests that mistake is highly likely, especially if the silent focal animal is partially hidden or is moving, and the second (caller) is nearby but hidden perfectly from the researcher (by grass, bush etc.). This is critically important for the results on individuality of the alarm calls.

ACCEPTED. We added the following part as follows (Lines 112-114):

“We included in the final analysis only calls of those individuals in whom it was possible to observe communication behavior, when the opening of the mouth during the calling could be seen.”

Lines 95-96 “The Animal Care and Use Committee of the Czech Ministry of the Environment approved this research.”

Please transfer this sentence in the Ethics section.

CORRECTED

Line 102 “Whistles”

Change “Whistles” to “Whistle alarms” throughout. The authors should indicate here that long-tailed ground squirrels produce two types of alarm calls, the whistle alarm and the wideband alarm, and refer to the respective figures with spectrograms.

ACCEPTED. We moved this sentence. (Lines 118-120)

Line 103 We analysed whistles of the highest quality.

Please provide the number of selected whistle alarms (as you did below for the wideband alarms). Please provide the criteria of the highest quality (for example, calls with high signal-to noise ratio, not broken with wind, not overlapped with alien calls or noises etc.).

ACCEPTED. We added the following sentence (Lines 123-125):

“We included only calls with a high signal-to noise ratio, not disturbed with a wind and not overlapping with calls of other individuals or background noises.”

Line 103. Please indicate in detail, how many whistle alarms were selected for analyses. From how many individuals. How many calls per animal (Mean, SD). How calls were selected within individual (from one of from a few series, following each other or not).

ACCEPTED. We included the information as follows (Lines 123-127, 147-148):

“We analysed 269 whistle alarms of the highest quality (20.7 ± 7.4 calls, mean \pm SD). We included only calls with a high signal-to noise ratio, not disturbed with a wind and not overlapping with calls of other individuals or background noises. From these calls we selected up to 32 calls per individual. In the case of larger number of calls we selected them randomly across all recorded series.”

“We selected 591 wideband alarms (23.7 ± 6.0 calls) of the highest quality from 31 individuals.”

Lines 104 and further. software

Please provide the firm, headquarters city and country of producer, this is a common claim of all journals.

ACCEPTED. We added this information. (Lines 128-129)

Lines 106-107 Background noise was filtered out using both high-pass (1.3–2.9 kHz) and low-pass filters (5.2–7.7 kHz).

The authors should write here, why these frequency ranges were filtered out. Remains unclear, whether such filtering mode affected the important parts of the studied calls. From Table 1 follows, that maximum values of the start fundamental frequency (6.65 kHz) are within the zone which was filtered out.

ACCEPTED. We included the following explanation (Lines 131-135):

“When the presence of the background noise, it was filtered out using both high-pass (1.3–2.9 kHz) and low-pass filters (5.2–7.7 kHz). These filters show range both for the lower and upper frequency limits. When the presence of the surrounding noise we filtered that noise which was out of the signal, e.g. when fundamental frequency reached up to 6.65 kHz, we filtered out the frequency higher than 7 kHz. ”

Lines 107-115

This list of measured acoustic parameters is non-informative. Unclear what the authors indeed measured. What do you mean under “frequency”? The fundamental frequency? Or the automatically measured peak frequency (frequency of maximum amplitude)? The authors should obligatory provide the figure illustrating all measured acoustic variables.

ACCEPTED. We changed this part and added the figure (Lines 136-144):

“We measured 11 acoustic variables: Duration, five frequency parameters (Fig. 2) using automatic measurements at five regular intervals of fundamental frequency duration...”

Figure 2. Measured points on the spectrograms for calculation of acoustical parameters quantifying the fundamental frequency

Line 118 “We selected...”

Please provide criteria for this selection of wideband alarm calls for analysis. Please indicate in detail, how many wideband alarms was selected per animal (Mean, SD). How calls were selected within individual (from one or a few call series, following each other or not).

ACCEPTED. We changed and expanded this part as follows (Lines 147-151):

“Similarly to whistle alarms, we included only calls with a high signal-to noise ratio, not overlapped with other calls or sounds. Signals where there was no clear lower frequency limit of the signal due to overlap with low frequency noise were not selected. From selected calls we randomly selected up to 34 calls per individual across all recorded series.”

Lines 119-120 Why 1.6 kHz high-pass filtering was applied for filtration? What was frequency range of the calls? Were meaningful call part damaged due to this filtering? Judging by spectrograms presented on Figure 2, the authors cut-off the lower halves of calls and then only analyses the upper halves. This is very strange and demands a detailed explanation. Moreover, many values of acoustic variables indicated in Table2 are lower 1600 Hz. How it is possible?

ACCEPTED. We changed this part as follows (Lines 147-151):

“Similarly to whistle alarms, we included only calls with a high signal-to noise ratio, not overlapped with other calls or sounds. Signals where there was no clear lower frequency limit of the signal due to overlap with low frequency noise were not selected. From selected calls we randomly selected up to 34 calls per individual across all recorded series.”

Line 125 Reference to Table 2 appears in text earlier then the reference to Table 1. Please correct this.

CORRECTED

Line 124 We measured 22 parameters

How these 22 parameters were measured? Manually? Automatically?

ACCEPTED. We changed the sentence as follows (Lines 158-159):

“We automatically measured 22 parameters quantifying how the acoustic energy is spread across the frequency spectrum.....”

Lines 124-125, Table 2. This description of measured parameters in insufficient. Please provide a detailed description of parameters being measured and provide a figure with illustrative spectrogram showing the measured acoustic variables. The used acoustic variables are unclear and are incompatible with the acoustic variables used in other papers on the alarm calls of ground squirrels, what makes the results of MS incomparable with other published studies. Please, make the descriptions of the acoustic variables clear at least for reviewer - bioacoustic researcher.

For example, 1st Quartile Frequency – this is a part of call spectrum, covering 25% of the total call energy. However, Table 2 describes the 1st Quartile Frequency as "The frequency that divides the selection into two frequency intervals containing 25% and 75% of the energy in the signal". How Q1F differs from Q1FRel, which is determined as "The frequency that divides the selection into two frequency intervals containing 25% and 75% of the energy in the signal relative to the frequency range of the signal"?

"Maximum entropy calculated from each frame". What is frame and how many frames?

"The difference between the 5% and 95% frequencies". Bandwidth is the width of the frequency band of call spectrum. What you understand under "frequency"? How 1.6 kHz high-pass filtering affects the bandwidth?

Center Frequency "The frequency that divides the selection into two frequency intervals of equal energy". What frequency, fundamental or peak? Or this is the median of call power spectrum? etc.

ACCEPTED. We took acoustic parameter descriptions from the software manual.

1st Quartile Frequency

The frequency that divides the selection into two frequency intervals containing 25% and 75% of the energy in the selection. The computation of this measurement is similar to that of Center Frequency, except that the summed energy has to exceed 25% of the total energy instead of 50%. For the spectrogram slice view, the procedure is the same except that the summation over time is not necessary since selections in the slice view occupy only one time bin. Units: Hz.

3rd Quartile Frequency

The frequency that divides the selection into two frequency intervals containing 75% and 25% of the energy in the selection. The computation of this measurement is similar to that of Center Frequency, except that the summed energy has to exceed 75% of the total energy instead of 50%. For the spectrogram slice view, the procedure is the same except that the summation over time is not necessary since selections in the slice view occupy only one time bin. Units: Hz.

Aggregate Entropy

The aggregate entropy measures the disorder in a sound by analyzing the energy distribution within a selection. Higher entropy values correspond to greater disorder in the sound whereas a pure tone with energy in only one frequency bin would have zero entropy.

Average Entropy

The average entropy in a selection is calculated by finding the entropy for each frame in the selection and then taking the average of these values. Unlike the aggregate entropy which uses the total energy in a frequency bin over the full time span, the average entropy calculates an entropy value for each slice in time and then averages. As a result, the average entropy measurement describes the amount of disorder for a typical spectrum within the selection, whereas the aggregate entropy corresponds to the overall disorder in the sound.

Bandwidth 90%

The difference between the 5% and 95% frequencies. Units: Hz

Center Frequency

The frequency that divides the selection into two frequency intervals of equal energy.

Center Time

The point in time at which the selection is divided into two time intervals of equal energy.

Frequency 5%

The frequency that divides the selection into two frequency intervals containing 5% and 95% of the energy in the selection. The computation of this measurement is similar to that of Center Frequency, except that the summed energy has to exceed 5% of the total energy instead of 50%. For the spectrogram slice view, the procedure is the same except that the summation over time is not necessary since selections in the slice view occupy only one time bin. Units: Hz.

Frequency 95%

The frequency that divides the selection into two frequency intervals containing 95% and 5% of the energy in the selection. The computation of this measurement is similar to that of Center Frequency, except that the summed energy has to exceed 95% of the total energy instead of 50%. For the spectrogram slice view, the procedure is the same except that the summation over time is not necessary since selections in the slice view occupy only one time bin. Units: Hz

IQR (Inter-quartile Range) Bandwidth

The difference between the 1st and 3rd Quartile Frequencies.

Max Frequency/ Peak Frequency

The frequency at which Max Power/ Peak Power occurs within the selection. If Max Power/ Peak Power occurs at more than one time and/or frequency, the lowest frequency at Max Time at which Max Power/ Peak Power occurs. Units: Hz.

Time 5%

The point in time that divides the selection into two time intervals containing 5% and 95% of the energy in the selection. The computation of this measurement is similar to that of Center Time, except that the summed energy has to exceed 5% of the total energy instead of 50%. For the spectrogram slice view, the procedure is the same except that the summation over time is not necessary since selections in the slice view occupy only one time bin. Units: seconds.

Time 95%

The point in time that divides the selection into two time intervals containing 95% and 5% of the energy in the selection. The computation of this measurement is similar to that of Center Time, except that the summed energy has to exceed 95% of the total energy instead of 50%. For the spectrogram slice view, the procedure is the same except that the summation over time is not necessary since selections in the slice view occupy only one time bin. Units: seconds.

The following explanation was sent by the creator of the program:

"1st Quartile Frequency" identifies the frequency in Hertz. 1st Quartile Frequency Relative is the same frequency expressed as a proportion of the bandwidth in a given selection (*). For example, if a selection has a Low Freq of 2000 Hz, High Freq of 3000 Hz, and Q1 Freq calculated as 2250 Hz, you would expect that Q1 Freq Rel. to be approximately 0.250.

(*) Selection is the selected part of the file for analysis, in this case it defines the signal in the time axis and frequency spectrum (beginning and end of the signal, lowest and highest frequency)

"Maximum entropy calculated from each frame": The number of frames in each selection is reported in the "Length (frames)" measurement. For each selection the "Length (frames)" measurement for the spectrogram line will report the number of spectrogram frames within the selection, and the "Length (frames)" measurement for the waveform line will report the number of

digital samples within the selection.

"Center Frequency" - it is the median frequency in the sense that it is the frequency at which 50% of the sound energy in the selection is below and 50% is above. Center Frequency is often not equal to fundamental frequency or peak frequency. Center Frequency is calculated in the spectrogram view, not a single spectrum (except for selections that have a "Length (frame)" measurement equal to "1").

"Relative" means relative to selection bounds. Peak Time Relative expressed the time at which the spectrogram bin with the highest spectrogram level occurs, relative to the duration of the selection. Frequency 95% expresses the proportion of the bandwidth below which 95% of the sound energy in the selection occurs.

Line 128. Individuals with a lower number of calls ($n < 7$)
What calls, whistle alarms or wideband alarms?

CORRECTED. A minimum number of calls produced by individuals included in resulting DFA models were 9 calls, both whistles and wideband calls. We changed the sentence to ($n < 9$). (Line 162)

Line 128-134. The results of a DFA are dependent from the sample size. I would suggest to balance the data set by number of calls per individual, included in each DFA.

ACCEPTED. We changed DFA as follows (Lines 169-171):

“To avoid a potential bias in DFA results due to an unbalanced dataset we randomly selected 9 calls per each individual and such balanced datasets entered into each DFA (whistles alarms: $n = 108$ calls, $N = 12$ individuals, wideband alarms: $n = 207$ calls, $N = 23$ individuals).”

Line 137-138. Six acoustic variables (Call duration, Element duration, Start Frequency, End Frequency, Min Frequency and Max Frequency)

It is necessary to provide a detailed description of these acoustic variables. Call duration – was this the duration of a series of repetitive calls? Frequency – was this the fundamental or the peak frequency? How it was measured when the alarm call represented a series of repetitive calls? Without clear methods, results cannot be understood.

ACCEPTED. We changed it and added an explanatory figure (Lines 136-144):

“We measured 11 acoustic variables quantifying a fundamental frequency of single whistle call: Duration, five frequency parameters (Fig. 2) using automatic measurements at five regular intervals of fundamental frequency duration of one whistle call (Start Frequency, 2nd Frequency,....”

Figure 2. Measured points on the spectrograms for calculation of acoustical parameters quantifying the fundamental frequency

Results

Line 146 and further. Alarm call description

As you analysed evidently different call numbers from different individuals, would be better to calculate the average values of acoustic variable for each individual and then use them in the descriptive statistics in Table 1. Such sample will be more balanced for calculating the averages per species than data pooling, used in MS.

**Note: the descriptive statistics were calculated from the average values per each individual
We added it under Methods as well.**

Line 147 tonal whistles

Change to whistle alarms throughout, uniformly with wideband alarms.

CORRECTED.

Lines 148-149 “calls with a tonal frequency modulated structure formed of one syllable only”
I suggest re-writing to “one-syllable calls with a visible modulated fundamental frequency”

ACCEPTED (Line 175)

Line 149 mean \pm SD

Transfer this in Statistics section, e.g. “All means were indicated as mean \pm SD”.

ACCEPTED. We moved it under the Methods/Statistics (Line 172)

Line 156 the second type of alarm signal

Please avoid synonyms, moreover “signal” is incorrect term here. I suggest rewriting to “the second alarm call type,”

CORRECTED (Line 183)

Line 166 Tonal alarms (Whistles)

Do not use synonyms, change with whistle alarms

CORRECTED (Line 193)

Lines 167-168. We excluded the highly correlated 2nd Frequency from all subsequent analysis. In Methods (Lines 131-132) you wrote that all 6 acoustic variables of whistle alarms were included in DFA. Please correct.

ACCEPTED. This part has been changed based on the balanced dataset used in DFA.

Lines 167-179. The discrimination analyses considered 269 whistles from 13 individuals. The results of a DFA are dependent from the sample size. Please provide a table with DFA results, including the number of whistle alarms from each of the 13 individuals and percentages of correct assignment of calls to each individual. Probably, the low value of correct assignment for some individuals is related to misbalanced sample.

ACCEPTED. This part has been changed based on the balanced dataset used in DFA. Number of calls per each individual included in the resulting DFA was balanced (n =9). It is mentioned in the Methods/Statistics.

We added the following table showing the classification results (Table S2. Classification results for whistles).

%	1	100,0	,0	,0	,0	,0	,0	,0	,0	,0	,0	,0	,0	100,0
	2	,0	100,0	,0	,0	,0	,0	,0	,0	,0	,0	,0	,0	100,0
	3	,0	,0	88,9	,0	,0	,0	,0	11,1	,0	,0	,0	,0	100,0
	4	,0	11,1	,0	88,9	,0	,0	,0	,0	,0	,0	,0	,0	100,0
	5	,0	,0	,0	,0	100,0	,0	,0	,0	,0	,0	,0	,0	100,0
	6	,0	,0	,0	,0	,0	100,0	,0	,0	,0	,0	,0	,0	100,0
	8	,0	,0	,0	,0	,0	,0	77,8	11,1	,0	,0	,0	11,1	100,0
	9	,0	,0	,0	,0	,0	,0	,0	100,0	,0	,0	,0	,0	100,0
	10	,0	,0	,0	,0	,0	,0	,0	,0	77,8	11,1	,0	11,1	100,0
	11	,0	,0	,0	,0	,0	33,3	,0	11,1	,0	55,6	,0	,0	100,0
	12	,0	11,1	,0	,0	,0	,0	,0	,0	,0	,0	88,9	,0	100,0
	13	,0	,0	,0	,0	,0	,0	,0	,0	11,1	,0	,0	88,9	100,0

Lines 178-179. Such classification results were much higher in comparison to classification by chance (7.7%).

Classification by chance was calculated incorrectly. Please use either the method by Solow, 1990 (Ecology, 71:2379-2382), or by Mundry, Sommer, 2007 (Animal Behaviour, 74:965-976)

Note: After discussion with the author of this work, we are not able to conduct suggested permuted DFA (pDFA) because it is related to two factorial data (e.g. individual and species, etc.) in crossed or nested design and our calculation is appropriate. We discussed the same problem during our previous work (Špinka et al. 2019)

Špinka et al. (2019). "Early vocal ontogeny in a polytocous mammal: no evidence of social learning among sibling piglets, *Sus scrofa*." *Animal Behaviour* 151: 9-19.

Lines 182-183. The model contained ten acoustic variables.

Again, earlier in Methods (Line 130) is indicated that 16 acoustic variables was included in analysis. Please correct.

ACCEPTED. This part has been changed as well based on the balanced dataset used in DFA.

Lines 182-192. We entered 591 wideband alarms from 25 individuals into the following analysis. The results of a DFA are dependent from the sample size. Please provide a table with DFA results, providing the numbers of wideband alarms from each of the 25 individuals and percentages of correct assignment to each individual. Probably, the low value of correct assignment for some individuals is related to misbalanced sample.

ACCEPTED. This part has been changed based on the balanced dataset used in DFA. Number of calls per each individual included in the resulting DFA was balanced (n =9). It is mentioned in the Methods/Statistics.

We added the following table showing the classification results (Table S3. Classification results for wideband calls).

% 1	88,9	,0	,0	,0	,0	,0	,0	,0	11,1	,0	,0	,0	,0	,0	,0	,0	,0	,0	,0
2	,0	88,9	,0	,0	,0	,0	,0	,0	,0	,0	,0	,0	11,1	,0	,0	,0	,0	,0	,0
4	,0	,0	100,0	,0	,0	,0	,0	,0	,0	,0	,0	,0	,0	,0	,0	,0	,0	,0	,0
5	,0	,0	,0	55,6	,0	,0	,0	,0	11,1	,0	,0	,0	,0	,0	33,3	,0	,0	,0	,0
6	22,2	,0	11,1	,0	66,7	,0	,0	,0	,0	,0	,0	,0	,0	,0	,0	,0	,0	,0	,0
7	,0	,0	11,1	,0	,0	66,7	11,1	,0	11,1	,0	,0	,0	,0	,0	,0	,0	,0	,0	,0
8	,0	,0	,0	,0	,0	,0	88,9	,0	,0	,0	,0	,0	,0	11,1	,0	,0	,0	,0	,0
9	,0	,0	,0	11,1	11,1	,0	,0	55,6	11,1	11,1	,0	,0	,0	,0	,0	,0	,0	,0	,0
11	,0	,0	11,1	,0	,0	,0	,0	,0	66,7	,0	,0	,0	,0	,0	22,2	,0	,0	,0	,0
12	,0	,0	,0	,0	,0	,0	,0	,0	,0	,0	100,0	,0	,0	,0	,0	,0	,0	,0	,0
13	,0	,0	,0	,0	,0	,0	,0	,0	,0	,0	,0	100,0	,0	,0	,0	,0	,0	,0	,0
14	,0	,0	,0	,0	,0	,0	,0	,0	,0	,0	,0	,0	100,0	,0	,0	,0	,0	,0	,0
15	11,1	11,1	,0	,0	,0	,0	,0	,0	11,1	,0	11,1	,0	,0	55,6	,0	,0	,0	,0	,0
18	,0	,0	,0	,0	,0	,0	,0	,0	,0	,0	,0	,0	,0	100,0	,0	,0	,0	,0	,0
21	,0	,0	,0	,0	,0	,0	,0	,0	11,1	11,1	,0	,0	,0	,0	77,8	,0	,0	,0	,0
22	,0	,0	,0	,0	,0	,0	,0	,0	,0	,0	,0	,0	,0	,0	,0	100,0	,0	,0	,0
24	,0	,0	,0	,0	,0	,0	,0	,0	,0	,0	,0	,0	,0	,0	,0	,0	100,0	,0	,0
25	,0	,0	,0	,0	,0	,0	,0	,0	,0	,0	,0	,0	,0	,0	,0	,0	,0	100,0	100,0

Lines 191-192. These results were much higher than classification by chance (4%). Classification by chance was calculated incorrectly. Please use either the method by Solow, 1990 (Ecology, 71:2379-2382), or by Mundry, Sommer, 2007 (Animal Behaviour, 74:965-976)

Note: After discussion with the author of this work, we are not able to conduct suggested permuted DFA (pDFA) because it is related to two factorial data (e.g. individual and species, individual and sex, etc.) in crossed or nested design and our calculation is appropriate. We discussed the same problem during our previous work (Špinka et al. 2019).

Špinka et al. (2019). "Early vocal ontogeny in a polytocous mammal: no evidence of social learning among sibling piglets, *Sus scrofa*." *Animal Behaviour* 151: 9-19.

Line 194. Fig. 4

Reference to Fig 3 and description of this figure are lacking in the MS.

CORRECTED. This sentence has been deleted based on the balanced dataset used in DFA.

Line 200 and further. Species comparison

Unfortunately, this interesting section is written based on incomplete and unclear how obtained data. Accordingly, the results of this section have a negligible meaning. this section should be re-

worked and re-calculated perfectly.

First, the authors should provide in Methods a detailed description of the used acoustic variables. For example, what frequency was measured (fundamental or peak?) in the wideband alarms of the long-tailed ground squirrel and how it was measured?

I think that the results of Fig 9 about the separate position of the wideband alarm of the long-tailed ground squirrel directly result from the incorrect measurements of “frequency”.

Second, remains unclear, from where and how the data introduced by the authors in cluster-analysis were obtained. Were they taken from the papers? Or from the own measurements of calls taken from web-sites? Of how many calls and from how many individuals?

Third, interspecies comparisons of the alarm calls of ground squirrels (Fig. 9 and Additional file 1) are evidently incomplete. Alarms of a few species available from literature (and even presented on Figs 5-8, but not included in Fig 9) should be included.

Data of the cue papers on the Arctic ground squirrel are lacking, although they are critically important for comparison of Eurasian and North American ground squirrels (Melchior, 1971, *Oecologia*, 7:184–190; Nikolskii, 1979, 1984).

The Arctic ground squirrel lives in tundra on both sides of Bering Strait and produces both the whistle and wideband alarms, very similar with the respective call types of the long-tailed ground squirrel. Without inclusion of these data, the picture of uniqueness of alarm calls of long-tailed ground squirrel is distorted.

The missing in MS references on the alarm calls of ground squirrels are available in Matrosova et al. 2012 (cited in MS) and in a new review on vocalizations of ground squirrels (Newar, Bowman, 2020, *Front. Ecol. Evol.*, 2020, v. 8: 00193. doi: 10.3389/fevo.2020.00193), especially in Supplementary to this review.

According to suggestion of the second reviewer we deleted this part (interspecies comparison) as it was not possible to compare the same acoustic parameters.

Lines 203-205 These calls represent tonal calls: monosyllabic whistles (A, D, E), two-syllable whistles (B, C), and multisyllabic chirps (F) and churrs (G).

Whistles, chirps, churrs: this terminology is redundant and only confuses the reader. I suggest re-writing to: “These alarm calls are all tonal calls: monosyllabic (A, D, E), two-syllabic (B, C), and multisyllabic (F, G).”

This part has been deleted.

Lines 205-208 “Calls of North American species (Fig. 6) also represent monosyllabic whistles (A, B, C), two-syllable whistles (D), and multisyllabic chirps (E, F). Similarly, calls of the genera *Otospermophilus* (Fig. 7) and *Ammospermophilus* represent multisyllabic signals (Fig. 8).”

Again, I suggest re-writing to: “Calls of North American species of the genus *Urocitellus* (Fig. 6) are also tonal calls: monosyllabic (A, B, C), two-syllabic (D), and multisyllabic (E, F). Similarly, the alarm calls of the genera *Otospermophilus* (Fig. 7) and *Ammospermophilus* (Fig. 8) are the multisyllabic calls.”

This part has been deleted.

Line 217 add “alarms” after three-syllable

CORRECTED

Line 218 add “alarm” before “calls”

CORRECTED

Discussion

Line 233

Delete “representing a whistle sound,”

CORRECTED

Line 235-236 which has no analogue in any other Eurasian ground squirrel species

It is incorrect, they are present in the Arctic ground squirrel living on Chukotka.

ACCEPTED. We changed the sentence as follows (Lines 233-235):

“...a wideband alarm call representing a different design of alarm signal (produced by long-tailed ground squirrel and arctic ground squirrel) has no analogue in any other Eurasian ground squirrel species.”

Line 238-240 We analysed the data based on published results, spectrograms and sounds to expand on the previous comparison of alarms produced by Eurasian and North American ground squirrels conducted by Matrosova et al. [8].

So is unclear why the authors did not include in analysis the listed by Matrosova et al. 2012 species of ground squirrels, for which literature data are available.

CORRECTED. We deleted the sentence.

Line 241

Delete “(whistles)” **CORRECTED**

Lines 250-251 “The wideband alarm, which we described in the long-tailed ground squirrel, represents a singly produced signal.”

I do not understand the sense of this sentence.

ACCEPTED. We deleted this sentence.

Lines 251-254. Such a call category of wideband alarms was not found either in our comparison or in the study of Matrosova et al. [8]. We suggest here that the long-tailed ground squirrel produces a wideband alarm signal which is something unique among Eurasian and North American species. It is incorrect. Matrosova et al. 2012 (Table 5) indicates that wideband alarm signal was described for the Arctic ground squirrel (Melchior, 1971; Nikolskii, 1984) and for the Californian ground squirrel (Owings, Virginia, 1978).

ACCEPTED. We deleted this part.

Line 253

Delete “signal” and “unique”

DELETED

Line 256

Change “collected in” to “created”

CORRECTED

Line 270 [see 5],

This reference is incorrect. Please replace it with Melchior, 1971; Owings, Virginia, 1978 (see Table 1 in this paper); Nikolskii, 1984

CORRECTED

Line 276

Change “signals” with “calls”

CORRECTED

Lines 281-282 “Tonal alarms could be assigned correctly to the other species of the genus *Urocitellus* (*U. beldingi*) with 70% success [25].”

I do not understand the sense of this sentence, please re-write.

CORRECTED. We deleted the sentence.

Lines 289-290. a wideband alarm, which has no similar analogue among ground squirrels either from Eurasia or North America.

It is incorrect; it was described for the Arctic ground squirrel.

ACCEPTED. We changed it as follows (Lines 233-235):

“...a wideband alarm call representing a different design of alarm signal (produced by long-tailed ground squirrel and arctic ground squirrel) has no analogue in any other Eurasian ground squirrel species.”

Lines 290-291. “Such findings indicate a phylogenetic signal present across alarms of ground squirrels when long-tailed squirrels is phylogenetically”

Sense is unclear, please re-write and correct the typo of ground

This part has been deleted.

Lines 384-386 Delete

CORRECTED

Line 397. Scatterplot of tonal alarms (whistles) from 13 individuals.

Please change to Scatterplot on the basis of two first discriminant functions of DFA for whistle alarms from 13 individuals

CORRECTED. We created new figure combining scatterplots of DFA results for whistle alarms and wideband alarms. (Figure 3, line 399)

Line 399. Scatterplot of wideband alarms from 25 individuals.

Please change to Scatterplot on the basis of two first discriminant functions of DFA for wideband alarms from 25 individuals. Please provide a legend with ID number of animals on Fig 4, similarly to Fig 3.

CORRECTED. We created new figure combining scatterplots of DFA results for whistle alarms and wideband alarms. (Figure 3, line 399)

Additional file 1. *Spermophilus suslicus*

This species produces only the monosyllabic whistle alarms. The authors indicate the minimum frequency large than the maximum frequency. Reference to Schneiderova & Policht 2012 Naturwissenschaften is incorrect, it is on different species.

CORRECTED. This file is no longer included as its content is no longer relevant to the text of the manuscript.

Reviewer: 2

Comments to the Author(s)

General comments:

This is an interesting manuscript touches on important problem about the role of individuality in the alarm calls of mammals. Ground-dwelling sciurids are a traditional model group for zoological research in various aspects.

The strongest aspect of the study, in my opinion, is the assessment of intraspecific individual differences, both in tonal and broadband calls. It is this part of the work that should be emphasized and developed first. Early studies were carried out at the species level and did not account for individual differences.

However, there were places where the article was problematic and unclear. Below are the areas which I found problematic.

1. The main drawback of the work I see is the superficial study of the information available in the literature. The authors present as the main result of the work the presence of two types of alarm calls in the long-tailed ground squirrel, however, this fact has long been well described. The first descriptions were made by Alexander Nikol'skii (1979, 1984).

ACCEPTED. We included these references including others. We changed the concept of the work as we shifted our attention to individual variability instead of phylogenetic comparison.

Also, not all Eurasian species for which there are publications (*S. pygmeus*, *S. mayor*, *S. erythrognys*, *U. parryi*) were taken for interspecies comparison. The absence of *U. parryi* greatly weakens the article, since the Arctic ground squirrel also live in Asia and belong to long-tailed ground squirrels.

ACCEPTED. We agree, but we did not have any data from *U. parryi*. In the end we avoided the phylogenetic part due to methodology problems.

2. The results of the study of individual traits require the use of an equal or similar number of sounds from each animal, the authors do not indicate this important information. The DFA should include an equal number of calls from an individual to obtain reliable results. It is necessary to write in the methods clearly how many individuals produced both types of alarm calls, from how many - only tonal, how many - only wideband. If each animal produced only one type of alarm call (either tonal or wideband), this should also be written clearly.

ACCEPTED. We changed DFA as follows (see Methods, lines 169-171, 110-111):

“To avoid a potential bias in DFA results due to an unbalanced dataset we randomly selected 9 calls per each individual and such balanced datasets entered into each DFA (whistles alarms: n = 108 calls, N = 12 individuals, wideband alarms: n = 207 calls, N = 23 individuals).”

“Each recorded individual used only one type of call, either an alarm whistle or a wideband signal.”

It is also advisable to add information about the number of examined individuals and calls to the Abstract section.

ACCEPTED. We added the information to the Abstract section.

3. Phylogenetic approach to the species pattern of the cry of alarm is an interesting but methodologically difficult task. The different structure of two types of alarm calls makes comparison difficult. Combining tonal and wideband alarm calls can only be correct when using completely identical call parameters (such as note duration). Comparison of the frequency parameters selected by the authors is incorrect, since in wideband alarm calls one can speak of dominant frequencies, the maximum and minimum fundamental frequencies cannot be measured.

In addition, it is not clear how to correctly compare with different numbers of notes. Thus, the last section of the results and the dendrogram of call similarity needs to be seriously revised or removed.

ACCEPTED. We deleted the dendrogram. We agree that comparison of non-identical parameters is problematic.

Minor remarks

Line 73. Are all the individuals examined adults? What is the likelihood of sampling big juveniles in July since the animals were unmarked?

ACCEPTED. We included information about age in the Methods and Table S1. Recorded individuals (Supplementary material).

Line 108-109, line 132, lines 137-138. Please provide a clear and correct description of the parameters used. Describe separately what was measured and separately what was calculated from the measurements (e.g. Frequency Modulation parameters) or how the primary measurements were interpreted (such as Min Frequency and Max Frequency).

The number of parameters for wideband calls looks excessive. I advise you to remove uninformative parameters from the article.

ACCEPTED. We changed this part as follows and added an explaining figure (Lines 136-144):

“We measured 11 acoustic variables quantifying a fundamental frequency of single whistle call: Duration, five frequency parameters (Figure.....) using automatic measurements at five regular intervals of fundamental frequency duration of one whistle call (Start Frequency, 2nd Frequency, Central Frequency, 4th Frequency, End Frequency) that divided each whistle into four sections in which we measured frequency modulation as follows: 1st Frequency Modulation (Start Frequency minus 2nd Frequency), 2nd Frequency Modulation (2nd Frequency minus Central Frequency), 3rd Frequency Modulation (Central Frequency minus 4th Frequency) and 4th Frequency Modulation (4th Frequency minus End Frequency), and Frequency Range (Start Frequency minus End Frequency).”

Figure 2. Measured points on the spectrograms for calculation of acoustical parameters quantifying the fundamental frequency

We excluded part related to inter-species comparison.

Lines 130-131 «The remaining 16 variables were entered into Discrimination Function Analysis». Enter the table with the DFA data and the remaining parameters. You can add this data to Supplementary.

ACCEPTED. Description of acoustic parameters includes Table 2 Acoustic parameter description. Classification results includes Table S2 for whistles and Table S3 for wideband calls. Correlations of acoustical parameters includes Table S4.

Lines 108-115 Give the number of animals and the number of whistles here. How much from each individual?

ACCEPTED. We added the information as follows (Lines 123-124):

“We analysed 269 whistle alarms of the highest quality produced by 13 individuals (20.7 ± 7.4 calls, mean \pm SD).”

Line 135-142. It is necessary to describe in detail the method of sound processing for interspecies comparison. Here you should describe where the alarm calls came from, how much from each species, how the parameters were measured. How were the durations in alarm cries with different numbers of elements counted? It is impossible to measure the parameters of the fundamental frequency for wideband calls, therefore, it is incorrect to combine them with tonal calls and probably should be considered separately or excluded from the scheme.

I highly recommend introducing the table «Additional_file_1._A» into the main body of the article for a better understanding of the material. In this case, the names of acoustic parameters in the table should be brought in line with the text of the article. You should also enter in this Table the number of individuals analyzed for each species.

We deleted this part (interspecies and phylogenetic) as recommended.

line 167 Give the spread in the number of studied sounds from each individual (average, from-to).

ACCEPTED. We added the information as follows (Lines 123-124):

“We analysed 269 whistle alarms of the highest quality produced by 13 individuals (20.7 ± 7.4 calls, mean ± SD).”

Table S1. Recorded individuals gives number of calls per each individual.

Line 182. Please, write the number of sounds from each individual and enter in the appropriate place in the Methods how many individuals were excluded from the analysis (probably 31-25 = 6 individuals).

ACCEPTED. We added the information as follows (Lines 147-148):

“We selected 591 wideband alarms (23.7 ± 6.0 calls) of the highest quality from 25 individuals.”

Table S1. Recorded individuals gives number of calls per each individual.

Lines 212-213. Information about the source of the screams should be moved to the Methods section.

This part has been deleted.

Lines 235-236. The Eurasian species of ground squirrels have wideband calls similar in structure in their vocal repertoire, but they are not used as an alarm calls.

ACCEPTED. We changed this part.

Lines 266-268. The position of *S.suslicus* and *S.fulvus* on the dendrogram completely contradicts the phylogenetic reconstruction based on mtDNA. In addition, the alarm calls of *S.suslicus* are monosyllabic, and in the diagram it is grouped with multisyllabic species. On the whole, speculation about the phylogenetic signal of the alarm calls looks speculative.

ACCEPTED. We deleted more-problematic inter-species comparison.

Figures

1. On all spectrograms there is no “Y” axis signature - sign “Frequency (kHz)”.

CORRECTED.

2. For a good perception of the article material, I would like to see different types of alarm calls nearby. I advise you to reduce the number of illustrations by grouping them into complex figures.

CORRECTED. Illustrations were grouped.

3. In Figure 2, label the individual numbers consistently with Figure 1. You can combine them into one figure with two parts (A, B).

CORRECTED. We added the labels and combined Figure 1 and 2 into one Figure 1.

4. Figure 3-4 is reasonable to combine into one.

CORRECTED. Figure 3-4 were replaced in accordance with the new analysis by Figure 3.

5. Figure 9. Normal font and italic are important information here, but they are very hard to see. Why is there no *S. erythrogastrus* in Figure 9? Why is RIC (*Urocitellus richardsonii*) shown twice in the figure?

CORRECTED. This file is no longer included as its content is no longer relevant to the text of the manuscript. We excluded figures related to inter-species comparison (Fig.5-9).

Tables

Table 2 refers to the methods, and is referred to in the text earlier than Table 1, which contains the results. Change places and table numbers.

CORRECTED